# Sequence learning, prediction, and replay in networks of spiking neurons

**Younes Bouhadjar**[1,2,3]*, **Dirk J. Wouters**[4], **Markus Diesmann**[1,5], **Tom Tetzlaff**[1]

**1** Institute of Neuroscience and Medicine (INM-6), & Institute for Advanced Simulation (IAS-6), & JARA BRAIN Institute Structure-Function Relationships (INM-10), Jülich Research Centre, Jülich, Germany, **2** Peter Grünberg Institute (PGI-7,10), Jülich Research Centre and JARA, Jülich, Germany, **3** RWTH Aachen University, Aachen, Germany, **4** Institute of Electronic Materials (IWE 2) & JARA-FIT, RWTH Aachen University, Aachen, Germany, **5** Department of Physics, Faculty 1, & Department of Psychiatry, Psychotherapy, and Psychosomatics, Medical School, RWTH Aachen University, Aachen, Germany

* y.bouhadjar@fz-juelich.de

**Data Availability Statement:** The documented workflow and source code necessary to reproduce our findings are provided online at: https://doi.org/10.5281/zenodo.5578212.

**Funding:** This project was funded by the Helmholtz Association Initiative and Networking Fund (project number SO-092, Advanced Computing

## Abstract

Sequence learning, prediction and replay have been proposed to constitute the universal computations performed by the neocortex. The Hierarchical Temporal Memory (HTM) algorithm realizes these forms of computation. It learns sequences in an unsupervised and continuous manner using local learning rules, permits a context specific prediction of future sequence elements, and generates mismatch signals in case the predictions are not met. While the HTM algorithm accounts for a number of biological features such as topographic receptive fields, nonlinear dendritic processing, and sparse connectivity, it is based on abstract discrete-time neuron and synapse dynamics, as well as on plasticity mechanisms that can only partly be related to known biological mechanisms. Here, we devise a continuous-time implementation of the temporal-memory (TM) component of the HTM algorithm, which is based on a recurrent network of spiking neurons with biophysically interpretable variables and parameters. The model learns high-order sequences by means of a structural Hebbian synaptic plasticity mechanism supplemented with a rate-based homeostatic control. In combination with nonlinear dendritic input integration and local inhibitory feedback, this type of plasticity leads to the dynamic self-organization of narrow sequence-specific subnetworks. These subnetworks provide the substrate for a faithful propagation of sparse, synchronous activity, and, thereby, for a robust, context specific prediction of future sequence elements as well as for the autonomous replay of previously learned sequences. By strengthening the link to biology, our implementation facilitates the evaluation of the TM hypothesis based on experimentally accessible quantities. The continuous-time implementation of the TM algorithm permits, in particular, an investigation of the role of sequence timing for sequence learning, prediction and replay. We demonstrate this aspect by studying the effect of the sequence speed on the sequence learning performance and on the speed of autonomous sequence replay.

Architectures) [YB, DJW, MD, TT], and the European Union's Horizon 2020 Framework Programme for Research and Innovation under the Specific Grant Agreement No.∼785907 (Human Brain Project SGA2) [YB, MD, TT] and No.∼945539 (Human Brain Project SGA3) [YB, MD, TT]. Open access publication funded by the Deutsche Forschungsgemeinschaft (DFG, German Research Foundation, 491111487) [YB, MD, TT]. The funders had no role in study design, data collection and analysis, decision to publish, or preparation of the manuscript.

**Competing interests:** The authors have declared that no competing interests exist.

## Author summary

Essentially all data processed by mammals and many other living organisms is sequential. This holds true for all types of sensory input data as well as motor output activity. Being able to form memories of such sequential data, to predict future sequence elements, and to replay learned sequences is a necessary prerequisite for survival. It has been hypothesized that sequence learning, prediction and replay constitute the fundamental computations performed by the neocortex. The Hierarchical Temporal Memory (HTM) constitutes an abstract powerful algorithm implementing this form of computation and has been proposed to serve as a model of neocortical processing. In this study, we are reformulating this algorithm in terms of known biological ingredients and mechanisms to foster the verifiability of the HTM hypothesis based on electrophysiological and behavioral data. The proposed model learns continuously in an unsupervised manner by biologically plausible, local plasticity mechanisms, and successfully predicts and replays complex sequences. Apart from establishing contact to biology, the study sheds light on the mechanisms determining at what speed we can process sequences and provides an explanation of fast sequence replay observed in the hippocampus and in the neocortex.

## Introduction

Learning and processing sequences of events, objects, or percepts are fundamental computational building blocks of cognition [1–4]. Prediction of upcoming sequence elements, mismatch detection and sequence replay in response to a cue signal constitute central components of this form of processing. We are constantly making predictions about what we are going to hear, see, and feel next. We effortlessly detect surprising, non-anticipated events and adjust our behavior accordingly. Further, we manage to replay learned sequences, for example, when generating motor behavior, or recalling sequential memories. These forms of processing have been studied extensively in a number of experimental works on sensory processing [5, 6], motor production [7], and decision making [8].

The majority of existing biologically motivated models of sequence learning addresses sequence replay [9–12]. Sequence prediction and mismatch detection are rarely discussed. The Hierarchical Temporal Memory (HTM) [13] combines all three aspects: sequence prediction, mismatch detection and replay. Its Temporal Memory (TM) model [14] learns complex context dependent sequences in a continuous and unsupervised manner using local learning rules [15], and is robust against noise and failure in system components. Furthermore, it explains the functional role of dendritic action potentials (dAPs) and proposes a mechanism of how mismatch signals can be generated in cortical circuits [14]. Its capacity benefits from sparsity in the activity, and therefore provides a highly energy efficient sequence learning and prediction mechanism [16].

The original formulation of the TM model is based on abstract models of neurons and synapses with discrete-time dynamics. Moreover, the way the network forms synapses during learning is difficult to reconcile with biology. Here, we propose a continuous-time implementation of the TM model derived from known biological principles such as spiking neurons, dAPs, lateral inhibition, spike-timing-dependent structural plasticity, and homeostatic control of synapse growth. This model successfully learns, predicts and replays high-order sequences, where the prediction of the upcoming element is not only dependent on the current element, but also on the history. Bringing the model closer to biology allows for testing its hypotheses based on experimentally accessible quantities such as synaptic connectivity, synaptic currents, transmembrane potentials, or spike trains. Reformulating the model in terms of continuous-

time dynamics moreover enables us to address timing-related questions, such as the role of the sequence speed for the prediction performance and the replay speed.

The study is organized as follows: the Methods describe the task, the network model, and the performance measures. The Results illustrate how the interaction of the model's components gives rise to context dependent predictions and sequence replay, and evaluate the sequence processing speed and prediction performance. The Discussion finally compares the spiking TM model to other biologically motivated sequence learning models, summarizes limitations, and provides suggestions for future model extensions.

## Methods

In the following, we provide an overview of the task and the training protocol, the network model, and the task performance analysis. A detailed description of the model and parameter values can be found in Tables 1 and 2.

### Task and training protocol

In this study, we develop a neuronal architecture that can learn and process an ensemble of $S$ sequences $s_i = \{\zeta_{i,1}, \zeta_{i,2}, \ldots, \zeta_{i,C_i}\}$ of ordered discrete items $\zeta_{i,j}$ with $C_i \in \mathbb{N}^+$, $i \in [1, \ldots, S]$. The length of sequence $s_i$ is denoted by $C_i$. Throughout this study, the sequence elements $\zeta_{i,j} \in \{A, B, C, \ldots\}$ are represented by Latin characters, serving as placeholders for arbitrary discrete objects or percepts, such as images, numbers, words, musical notes, or movement primitives (Fig 1A). The order of the sequence elements within a given sequence represents the temporal order of item occurrence.

The tasks to be solved by the network consist of

  i). predicting subsequent sequence elements in response to the presentation of other elements,

 ii). detecting unanticipated stimuli and generating a mismatch signal if the prediction is not met, and

iii). autonomously replaying sequences in response to a cue signal after learning.

The architecture learns sequences in a continuous manner: the network is exposed to repeated presentations of a given ensemble of sequences (e.g., {A, D, B, E} and {F, D, B, C} in Fig 1B). In the *prediction mode* (task i) and ii)), there is no distinction between a "training" and a "testing" phase. At the beginning of the learning process, all presented sequence elements are unanticipated and do not lead to a prediction (diffuse shades in Fig 1B, left). As a consequence, the network generates mismatch signals (flash symbols in Fig 1B, left). After successful learning, the presentation of some sequence element leads to a prediction of the subsequent stimulus (colored arrows in Fig 1B). In case this subsequent stimulus does not match the prediction, the network generates a mismatch signal (red arrow and flash symbol in Fig 1B, right). The learning process is entirely unsupervised, i.e., the prediction performance does not affect the learning. As described in Sequence replay, the network can be configured into a *replay mode* where the network autonomously replays learned sequences in response to a cue signal (task iii)).

In general, the sequences in this study are "high-order" sequences, similar to those generated by a high-order Markov chain; the prediction of an upcoming sequence element requires accounting for not just the previous element, but for (parts of) the entire sequence history, i.e., the context. Sequences within a given set of training data can be partially overlapping; they may share certain elements or subsequences (such as in {A, D, B, E} and {F, D, B, C}). Similarly, the same sequence element (but not the first one, see Limitations and outlook) may occur multiple times within the same sequence (such as in {A, D, B, D}). Throughout this work, we use two sequence sets:

**Table 1. Description of the network model.** Parameter values are given in Table 2.

| Summary | |
|---|---|
| **Populations** | excitatory neurons ($\mathcal{E}$), inhibitory neurons ($\mathcal{I}$), external spike sources ($\mathcal{X}$); $\mathcal{E}$ and $\mathcal{I}$ composed of $M$ disjoint subpopulations $\mathcal{M}_k$ and $\mathcal{I}_k$ ($k = 1, \ldots, M$) |
| **Connectivity** | • sparse random connectivity between excitatory neurons (plastic) <br> • local recurrent connectivity between excitatory and inhibitory neurons (static) |
| **Neuron model** | • excitatory neurons: leaky integrate-and-fire (LIF) with nonlinear input integration (dendritic action potentials) <br> • inhibitory neurons: leaky integrate-and-fire (LIF) |
| **Synapse model** | exponential or alpha-shaped postsynaptic currents (PSCs) |
| **Plasticity** | homeostatic spike-timing dependent structural plasticity in excitatory-to-excitatory connections |

| Populations | | |
|---|---|---|
| **Name** | **Elements** | **Size** |
| $\mathcal{E} = \cup_{i=k}^{M} \mathcal{M}_k$ | excitatory (E) neurons | $N_\text{E}$ |
| $\mathcal{I} = \cup_{i=k}^{M} \mathcal{I}_k$ | inhibitory (I) neurons | $N_\text{I}$ |
| $\mathcal{M}_k$ | excitatory neurons in subpopulation $k$, $\mathcal{M}_k \cap \mathcal{M}_l = \emptyset$ ($\forall k \neq l \in [1, M]$) | $n_\text{E}$ |
| $\mathcal{I}_k$ | inhibitory neurons in subpopulation $k$, $\mathcal{I}_k \cap \mathcal{I}_l = \emptyset$ ($\forall k \neq l \in [1, M]$) | $n_\text{I}$ |
| $\mathcal{X} = \{x_1, \ldots, x_M\}$ | external spike sources | $M$ |

| Connectivity | | |
|---|---|---|
| **Source population** | **Target population** | **Pattern** |
| $\mathcal{E}$ | $\mathcal{E}$ | random; fixed in-degrees $K_i = K_\text{EE}$, delays $d_{ij} = d_\text{EE}$, synaptic time constants $\tau_{ij} = \tau_\text{EE}$; plastic weights $J_{ij} \in \{0, J_{\text{EE},ij}\}$ ($\forall i \in \mathcal{E}, \forall j \in \mathcal{E}$; "EE connections") |
| $\mathcal{M}_k$ | $\mathcal{I}_k$ | all-to-all; fixed delays $d_{ij} = d_\text{IE}$, synaptic time constants $\tau_{ij} = \tau_\text{IE}$, and weights $J_{ij} = J_\text{IE}$ ($\forall i \in \mathcal{I}_k, \forall j \in \mathcal{M}_k, \forall k \in [1, M]$; "IE connections") |
| $\mathcal{I}_k$ | $\mathcal{M}_k$ | all-to-all; fixed delays $d_{ij} = d_\text{EI}$, synaptic time constants $\tau_{ij} = \tau_\text{EI}$, and weights $J_{ij} = J_\text{EI}$ ($\forall i \in \mathcal{M}_k, \forall j \in \mathcal{I}_k, \forall k \in [1, M]$; "EI connections") |
| $\mathcal{I}_k$ | $\mathcal{I}_k$ | none ($\forall k \in [1, M]$; "II connections") |
| $\mathcal{X}_k = x_k$ | $\mathcal{M}_k$ | one-to-all; fixed delays $d_{ik} = d_\text{EX}$, synaptic time constants $\tau_{ij} = \tau_\text{EX}$, and weights $J_{ik} = J_\text{EX}$ ($\forall i \in \mathcal{M}_k, \forall k \in [1, M]$; "EX connections") |

no self-connections ("autapses"), no multiple connections ("multapses")

all unmentioned connections $\mathcal{M}_k \to \mathcal{I}_l, \mathcal{I}_k \to \mathcal{M}_l, \mathcal{I}_k \to \mathcal{I}_l, \mathcal{X}_k \to \mathcal{M}_l$ ($\forall k \neq l$) are absent

| Neuron and synapse | |
|---|---|
| **Neuron** | |
| **Type** | leaky integrate-and-fire (LIF) dynamics |
| **Description** | dynamics of membrane potential $V_i(t)$ and spiking activity $s_i(t)$ of neuron $i$: <br><br> • emission of the $k$th spike of neuron $i$ at time $t_i^k$ if <br><br> $$V_i(t_i^k) \geq \theta_i \qquad (9)$$ <br> with somatic spike threshold $\theta_i$ <br> • spike train: $s_i(t) = \sum_k \delta(t - t_i^k)$ <br> • reset and refractoriness: <br><br> $$V_i(t) = V_\text{r} \quad \forall k, \ \forall t \in (t_i^k, t_i^k + \tau_{\text{ref},i}]$$ <br> with refractory time $\tau_{\text{ref},i}$ and reset potential $V_\text{r}$ <br> • subthreshold dynamics: <br><br> $$\tau_{\text{m},i} \dot{V}_i(t) = -V_i(t) + R_{\text{m},i} I_i(t) \qquad (10)$$ <br> with membrane resistance $R_{\text{m},i} = \dfrac{\tau_{\text{m},i}}{C_{\text{m},i}}$, membrane time constant $\tau_{\text{m},i}$, and total synaptic input current $I_i(t)$ (see Synapse) <br> • excitatory neurons: $\tau_{\text{m},i} = \tau_{\text{m,E}}$, $C_{\text{m},i} = C_\text{m}$, $\theta_i = \theta_\text{E}$, $\tau_{\text{ref},i} = \tau_{\text{ref,E}}$ ($\forall i \in \mathcal{E}$) <br> • inhibitory neurons: $\tau_{\text{m},i} = \tau_{\text{m,I}}$, $C_{\text{m},i} = C_\text{m}$, $\theta_i = \theta_\text{I}$, $\tau_{\text{ref},i} = \tau_{\text{ref,I}}$ ($\forall i \in \mathcal{I}$) |

(*Continued*)

| Synapse | |
|---|---|
| **Type** | exponential or alpha-shaped postsynaptic currents (PSCs) |
| **Description** | • total synaptic input currents: |

$$\text{excitatory neurons}: \quad I_i(t) \quad = I_{\text{ED},i}(t) + I_{\text{EX},i}(t) + I_{\text{EI},i}(t), \quad \forall i \in \mathcal{E}$$

$$\text{inhibitory neurons}: \quad I_i(t) \quad = I_{\text{IE},i}(t), \quad \forall i \in \mathcal{I} \tag{11}$$

with dendritic, external, inhibitory and excitatory input currents $I_{\text{ED},i}(t)$, $I_{\text{EX},i}(t)$, $I_{\text{EI},i}(t)$, $I_{\text{IE},i}(t)$ evolving according to

$$I_{\text{ED},i}(t) = \sum_{j \in \mathcal{E}} (\alpha_{ij} * s_j)(t - d_{ij}) \tag{12}$$

with $\alpha_{ij}(t) = J_{ij} \dfrac{e}{\tau_{\text{ED}}} t e^{-t/\tau_{\text{ED}}} \Theta(t)$ and $\Theta(t) = \begin{cases} 1 & t \geq 0 \\ 0 & \text{else} \end{cases}$,

$$\tau_{\text{EX}} \dot{I}_{\text{EX},i} = -I_{\text{EX},i}(t) + \sum_{j \in \mathcal{X}} J_{ij} s_j(t - d_{ij}), \tag{13}$$

$$\tau_{\text{EI}} \dot{I}_{\text{EI},i} = -I_{\text{EI},i}(t) + \sum_{j \in \mathcal{I}} J_{ij} s_j(t - d_{ij}), \tag{14}$$

$$\tau_{\text{IE}} \dot{I}_{\text{IE},i} = -I_{\text{IE},i}(t) + \sum_{j \in \mathcal{E}} J_{ij} s_j(t - d_{ij}) \tag{15}$$

with $\tau_{\text{EX}}$, $\tau_{\text{EI}}$, and $\tau_{\text{IE}}$ synaptic time constants of EX, EI, and IE connections, respectively, and $J_{ij}$ the synaptic weight

• suprathreshold dynamics of dendritic currents (dAP generation):

 • emission of $k$th dAP of neuron $i$ at time $t^k_{\text{dAP},i}$ if $I_{\text{ED},i}(t^k_{\text{dAP},i}) \geq \theta_{\text{dAP}}$

 • dAP current plateau:

$$I_{\text{ED},i}(t) = I_{\text{dAP}} \quad \forall k, \quad \forall t \in (t^k_{\text{dAP},i}, t^k_{\text{dAP},i} + \tau_{\text{dAP}}) \tag{16}$$

 with dAP current plateau amplitude $I_{\text{dAP}}$, dAP current duration $\tau_{\text{dAP}}$, and dAP activation threshold $\theta_{\text{dAP}}$

 • reset: $I_{\text{ED},i}(t^k_{\text{dAP},i} + \tau_{\text{dAP}}) = 0 \ (\forall k)$

 • reset and refractoriness in response to emission of $l$th somatic spike of neuron $i$ at time $t^l_i$: $I_{\text{ED},i}(t) = 0 \quad \forall l, \quad \forall t \in (t^l_i, t^l_i + \tau_{\text{ref},i})$

| Plasticity | |
|---|---|
| **Type** | spike-timing dependent structural plasticity and dAP-rate homeostasis |
| **EE synapses** | • dynamics of synaptic permanence $P_{ij}(t)$ (synapse maturity): |

$$\begin{aligned} P_{\text{max}}^{-1} \frac{dP_{ij}}{dt} &= \lambda_+ \sum_{\{t^*_i\}'} x_j(t) \delta(t - [t^*_i + d_{\text{EE}}]) I(t^*_i, \Delta t_{\text{min}}, \Delta t_{\text{max}}) + \\ &\quad -\lambda_- \sum_{\{t^*_j\}} y_i \delta(t - t^*_j) + \\ &\quad +\lambda_{\text{h}} \sum_{\{t^*_i\}'} (z^* - z_i(t)) \delta(t - t^*_i) I(t^*_i, \Delta t_{\text{min}}, \Delta t_{\text{max}}) \end{aligned}$$

with

• list of presynaptic spike times $\{t^*_j\}$,

• list of postsynaptic spike times $\{t^*_i\}' = \{t^*_i | \forall t^*_j : t^*_i - t^*_j + d_{\text{EE}} \geq \Delta t_{\text{min}}\}$,

• indicator function

$$I(t^*_i, \Delta t_{\text{min}}, \Delta t_{\text{max}}) = \sum_{\{t^*_j\}} R(t^*_i - t^*_j + d_{\text{EE}}) \text{ with } R(\tau) = \begin{cases} 1 & \Delta t_{\text{min}} < \tau < \Delta t_{\text{max}} \\ 0 & \text{else}, \end{cases} \tag{17}$$

• maximum permanence $P_{\text{max}}$, potentiation and depression rates $\lambda_+$, $\lambda_-$, homeostasis rate $\lambda_{\text{h}}$, delay $d_{\text{EE}}$, depression decrement $y_i$, minimum $\Delta t_{\text{min}}$ and maximum $\Delta t_{\text{max}}$ time lags between pairs of pre- and postsynaptic spikes at which synapses are potentiated,

• spike trace $x_j(t)$ of presynaptic neuron $j$, evolving according to

$$\frac{dx_j}{dt} = -\tau_+^{-1} x_j(t) + \sum_{t^*_j} \delta(t - t^*_j)$$

with presynaptic spike times $t^*_j$ and potentiation time constant $\tau_+$,

• dAP trace $z_i(t)$ of postsynaptic neuron $i$, evolving according to

$$\frac{dz_i}{dt} = -\tau_{\text{h}}^{-1} z_i(t) + \sum_k \delta(t - t^k_{\text{dAP},i})$$

with onset time $t^k_{\text{dAP},i}$ of the $k$th dAP, homeostasis time constant $\tau_{\text{h}}$, and

• target dAP activity $z^*$

*(Continued)*

 

- dynamics of synaptic weights $J_{EE,ij}$:

$$J_{EE,ij}(t) = \begin{cases} W & \text{if } P_{ij}(t) \geq \theta_P & \text{(mature synapse)} \\ 0 & \text{if } P_{ij}(t) < \theta_P & \text{(immature synapse)} \end{cases}$$

with weight of mature EE connections $W$ and synapse maturity threshold $\theta_P$

(for an algorithmic implementation of the plasticity dynamics, see S1 Algorithm)

| all other synapses | non-plastic |
|---|---|

**Input**

- prediction mode
  - repetitive stimulation with the same set $\mathcal{S} = \{s_1, \ldots, s_S\}$ of sequences $s_i = \{\zeta_{i,1}, \zeta_{i,2}, \ldots, \zeta_{i,C_i}\}$ of ordered discrete items $\zeta_{i,j}$ with number of sequences $S$ and length $C_i$ of $i$th sequence
  - presentation of sequence element $\zeta_{i,j}$ at time $t_{i,j}$ modeled by single spike $x_k(t) = \delta(t - t_{i,j})$, generated by the corresponding external source $x_k$
  - inter-stimulus interval $\Delta T = t_{i,j+1} - t_{i,j}$ between subsequent sequence elements $\zeta_{i,j}$ and $\zeta_{i,j+1}$ within a sequence $s_i$
  - inter-sequence time interval $\Delta T_{seq} = t_{i+1,1} - t_{i,C_i}$ between subsequent sequences $s_i$ and $s_{i+1}$
  - example sequence sets:
    - sequence set I: $\mathcal{S} = \{\{A, D, B, E\}, \{F, D, B, C\}\}$
    - sequence set II: $\mathcal{S} = \{\{E, N, D, I, J\}, \{L, N, D, I, K\}, \{G, J, M, C, N\}, \{F, J, M, C, I\}, \{B, C, K, H, I\}, \{A, C, K, H, F\}\}$
- replay mode
  - presentation of a cue encoding for first sequence elements $\zeta_{i,1}$ at $t_{i,1}$
  - inter-cue time interval $\Delta T_{cue} = t_{i+1,1} - t_{i,1}$ between subsequent cues $\zeta_{i,1}$ and $\zeta_{i+1,1}$

**Output**

- somatic spike times $\{t_i^k | \forall i \in \mathcal{E}, k = 1, 2, \ldots\}$
- dendritic currents $I_{ED,i}(t)$ $(\forall i \in \mathcal{E})$

**Initial conditions and network realizations**

- membrane potentials: $V_i(0) = V_r$ $(\forall i \in \mathcal{E} \cup \mathcal{I})$
- dendritic currents: $I_{ED,i}(0) = 0$ $(\forall i \in \mathcal{E})$
- external currents: $I_{EX,i}(0) = 0$ $(\forall i \in \mathcal{E})$
- inhibitory currents: $I_{EI,i}(0) = 0$ $(\forall i \in \mathcal{E})$
- excitatory currents: $I_{IE,i}(0) = 0$ $(\forall i \in \mathcal{I})$
- synaptic permanences: $P_{ij}(0) = P_{min,ij}$ with $P_{min,ij} \sim \mathcal{U}(P_{0,min}, P_{0,max})$ $(\forall i, j \in \mathcal{E})$
- synaptic weights: $J_{EE,ij}(0) = 0$ $(\forall i, j \in \mathcal{E})$
- spike traces: $x_i(0) = 0$ $(\forall i \in \mathcal{E})$
- dAP traces: $z_i(0) = 0$ $(\forall i \in \mathcal{E})$
- potential connectivity and initial permanences randomly and independently drawn for each network realization

**Simulation details**

- network simulations performed in NEST [34] version 3.0 [35]
- definition of excitatory neuron model using NESTML [36, 37]
- synchronous update using exact integration of system dynamics on discrete-time grid with step size $\Delta t$ [38]
- source code underlying this study: https://doi.org/10.5281/zenodo.5578212

**Sequence set I.** For an illustration of the learning process and the network dynamics in the prediction (section Sequence learning and prediction) and in the replay mode (section Sequence replay), as well as for the investigation of the sequence processing speed (section Dependence of prediction performance on the sequence speed), we start with a simple set of two partially overlapping sequences $s_1 = \{A, D, B, E\}$ and $s_2 = \{F, D, B, C\}$ (see Fig 1B).

**Sequence set II.** For a more rigorous evaluation of the sequence prediction performance (section Prediction performance), we consider a set of $S = 6$ high-order sequences: $s_1 = \{E, N, D, I, J\}$, $s_2 = \{L, N, D, I, K\}$, $s_3 = \{G, J, M, C, N\}$, $s_4 = \{F, J, M, C, I\}$, $s_5 = \{B, C, K, H, I\}$, $s_6 = \{A, C, K, H, F\}$,

 

**Table 2. Model and simulation parameters.** Parameters derived from other parameters are marked in gray. Bold numbers depict default values.

| Name | Value | Description |
|---|---|---|
| **Network** | | |
| $N_E$ | 2100 | total number of excitatory neurons |
| $N_I$ | 14 | total number of inhibitory neurons |
| $M$ | $A=14$ | number of excitatory subpopulations (= number of external spike sources) |
| $n_E$ | $N_E/M=150$ | number of excitatory neurons per subpopulation |
| $n_I$ | $N_I/M=1$ | number of inhibitory neurons per subpopulation |
| $\rho$ | 20 | (target) number of active neurons per subpopulation after learning = minimal number of coincident excitatory inputs required to trigger a spike in postsynaptic inhibitory neurons |
| **(Potential) Connectivity** | | |
| $K_{EE}$ | 420 | number of excitatory inputs per excitatory neuron (EE in-degree) |
| $p$ | $K_{EE}/N_E= 0.2$ | probability of potential (excitatory) connections |
| $K_{EI}$ | $n_I = 1$ | number of inhibitory inputs per excitatory neuron (EI in-degree) |
| $K_{IE}$ | $n_E$ | number of excitatory inputs per inhibitory neuron (IE in-degree) |
| $K_{II}$ | 0 | number of inhibitory inputs per inhibitory neuron (II in-degree) |
| **Excitatory neurons** | | |
| $\tau_{m,E}$ | 10 ms | membrane time constant |
| $\tau_{ref,E}$ | 10 ms | absolute refractory period |
| $C_m$ | 250 pF | membrane capacity |
| $V_r$ | 0.0 mV | reset potential |
| $\theta_E$ | 20 mV (predictive mode), 5 mV (replay mode) | somatic spike threshold |
| $I_{dAP}$ | 200 pA | dAP current plateau amplitude |
| $\tau_{dAP}$ | 60 ms | dAP duration |
| $\theta_{dAP}$ | 59 pA (predictive mode), 41.3 pA (replay mode) | dAP threshold |
| **Inhibitory neurons** | | |
| $\tau_{m,I}$ | 5 ms | membrane time constant |
| $\tau_{ref,I}$ | 2 ms | absolute refractory period |
| $C_m$ | 250 pF | membrane capacity |
| $V_r$ | 0.0 mV | reset potential |
| $\theta_I$ | 15 mV | spike threshold |
| **Synapse** | | |
| $\gamma$ | 5 | number co-active presynaptic neurons required to trigger a dAP in the postsynaptic neuron |
| $W$ | 12.98 pA | weight of mature EE connections (EPSC amplitude) |
| $\tilde{J}_{IE}$ | 0.9 mV (predictive mode), 0.12 mV (replay mode) | weight of IE connections (EPSP amplitude) |
| $J_{IE}$ | 581.19 pA (predictive mode), 77.49 pA (replay mode) | weight of IE connections (EPSC amplitude) |
| $\tilde{J}_{EI}$ | −40 mV | weight of EI connections (IPSP amplitude) |
| $J_{EI}$ | −12915.49 pA | weight of EI connections (IPSC amplitude) |
| $\tilde{J}_{EX}$ | 22 mV | weight of EX connections (EPSP amplitude) |
| $J_{EX}$ | 4112.20 pA | weight of EX connections (EPSC amplitude) |
| $\tau_{EE}$ | 5 ms | synaptic time constant of EE connections |
| $\tau_{IE}$ | 0.5 ms | synaptic time constant of IE connections |
| $\tau_{EI}$ | 1 ms | synaptic time constant of EI connections |
| $\tau_{EX}$ | 2 ms | synaptic time constant of EX connection |
| $d_{EE}$ | 2 ms | delay of EE connections (dendritic) |
| $d_{IE}$ | 0.1 ms | delay of IE connections |
| $d_{EI}$ | {**0.1**, 0.2} ms | delay of EI connections (non-default value used in Figs 10 and 11) |
| $d_{EX}$ | 0.1 ms | delay of EX connections |

*(Continued)*

**Table 2.** (Continued)

| Name | Value | Description |
|------|-------|-------------|
| | | **Plasticity** |
| $\lambda_+$ | 0.08 (sequence set I), 0.28 (sequence set II) | potentiation rate |
| $\lambda_-$ | 0.0015 (sequence set I), 0.0061 (sequence set II) | depression rate |
| $\theta_P$ | 20 | synapse maturity threshold |
| $P_{\text{min},ij}$ | $\sim \mathcal{U}(P_{0,\text{min}}, P_{0,\text{max}})$ | minimum permanence |
| $P_{\text{max}}$ | 20 | maximum permanence |
| $P_{0,\text{min}}$ | 0 | minimal initial permanence |
| $P_{0,\text{max}}$ | 8 | maximal initial permanence |
| $\tau_+$ | 20 ms | potentiation time constant |
| $z^*$ | 1 | target dAP activity |
| $\lambda_h$ | 0.014 (sequence set I), 0.024 (sequence set II) | homeostasis rate |
| $\tau_h$ | 440 ms (sequence set I), 1560 ms (sequence set II) | homeostasis time constant |
| $y_i$ | 1 | depression decrement |
| $\Delta t_{\text{min}}$ | 4 ms | minimum time lag between pairs of pre- and postsynaptic spikes at which synapses are potentiated |
| $\Delta t_{\text{max}}$ | $2\Delta T$ | maximum time lag between pairs of pre- and postsynaptic spikes at which synapses are potentiated |
| | | **Input** |
| $L$ | 1 | number of subpopulations per sequence element = number of target subpopulations per spike source |
| $S$ | 2 (sequence set I), 6 (sequence set II) | number of sequences per set |
| $C$ | 4 (sequence set I), 5 (sequence set II) | number of elements per sequence |
| $A$ | 14 | alphabet length (total number of distinct sequence elements) |
| $\Delta T$ | $\{2,\ldots,\mathbf{40},\ldots,90\}$ ms | inter-stimulus interval |
| $\Delta T_{\text{seq}}$ | $\max(2.5\Delta T, \tau_{\text{dAP}})$ | inter-sequence interval |
| $\Delta T_{\text{cue}}$ | 80 ms | inter-cue interval |
| | | **Simulation** |
| $\Delta t$ | 0.1 ms | time resolution |
| $K$ | $\{\mathbf{80}, 100\}$ | number of training episodes |

each consisting of $C = 5$ elements. The complexity of this sequence ensemble is comparable to the one used in [14], but is more demanding in terms of the high-order context dependence.

Results for two additional sequence sets are summarized in the Supporting information. The set used in S2 Fig is composed of sequences with recurring first elements. In S3 Fig, we show results for longer sequences with a larger number of overlapping elements.

## Network model

**Algorithmic requirements.** To solve the tasks outlined in Task and training protocol, the network model needs to implement a number of algorithmic components. Here, we provide an overview of these components and their corresponding implementations:

- Learning and storage of sequences: in both the original and our model, sequences are represented by specific subnetworks embedded into the recurrent network. During the learning process, these subnetworks are carved out in an unsupervised manner by a form of structural Hebbian plasticity.

**A**

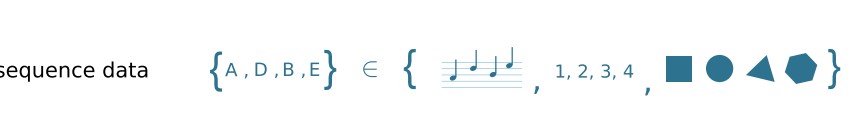

**B**

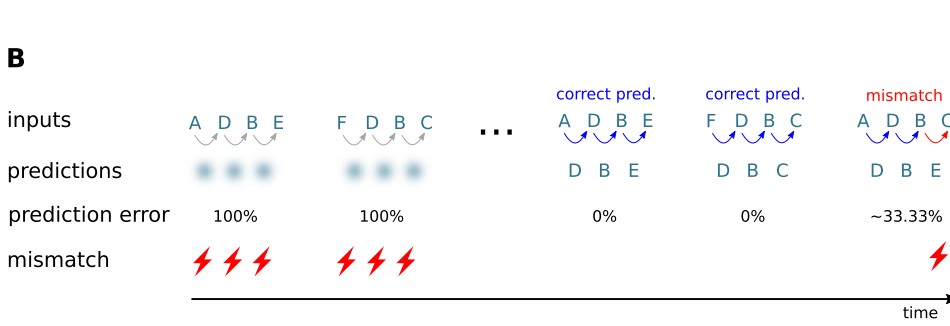

**Fig 1. Sketch of the task and the learning protocol. A)** The neuronal network model developed in this study learns and processes sequences of ordered discrete elements, here represented by characters "A", "B", "C", . . .. Sequence elements may constitute arbitrary discrete items, such as musical notes, numbers, or images. The order of sequence elements represents the temporal order of item occurrence. **B)** After repeated, consistent presentation of sets of high-order sequences, i.e., sequences with overlapping characters (here, {A, D, B, E} and {F, D, B, C}), the model learns to predict subsequent elements in response to the presentation of other elements (blue arrows) and to detect unanticipated elements by generating a mismatch signal if the prediction is not met (red arrows and flash symbols). The learning process is continuous and unsupervised. At the beginning of the learning process, all presented elements are unanticipated and hence trigger the generation of a mismatch signal. The learning progress is monitored and quantified by the prediction error (see Task performance measures).

- Context specificity: in our model, learning of high-order sequences is enabled by a sparse, random potential connectivity, and by a homeostatic regulation of synaptic growth.

- Generation of predictions: neurons are equipped with a predictive state, implemented by a nonlinear synaptic integration mimicking the generation of dendritic action potentials (dAPs).

- Mismatch detection: only few neurons become active if a prediction matches the stimulus. In our model, this sparsity is realized by a winner-take-all (WTA) dynamics implemented in the form of inhibitory feedback. In case of non-anticipated stimuli, the WTA dynamics cannot step in, thereby leading to a non-sparse activation of larger neuron populations.

- Sequence replay: autonomous replay of learned sequences in response to a cue signal is enabled by increasing neuronal excitability.

In the following paragraphs, the implementations of these components and the differences between the original and our model are explained in more detail.

**Network structure.** The network consists of a population $\mathcal{E}$ of $N_E$ excitatory ("E") and a population $\mathcal{I}$ of $N_I$ inhibitory ("I") neurons. The neurons in $\mathcal{E}$ are randomly and recurrently connected, such that each neuron in $\mathcal{E}$ receives $K_{EE}$ excitatory inputs from other randomly chosen neurons in $\mathcal{E}$. Note that these "EE" connections are potential connections in the sense that they can be either "mature" ("effective") or "immature". Immature connections have no effect on target neurons (see below). In the neocortex, the degree of potential connectivity depends on the distance between the neurons [17]. It can reach probabilities as high as 90% for neighboring neurons, and decays to 0% for neurons that are farther apart. In this work, the connection probability is chosen such that the connectivity is sufficiently dense, allowing for the formation of specific subnetworks, and sufficiently sparse for increasing the network capacity (see paragraph "Constraints on potential connectivity" below). The excitatory population $\mathcal{E}$ is subdivided into $M$ non-overlapping subpopulations $\mathcal{M}_1, \ldots, \mathcal{M}_M$, each of them

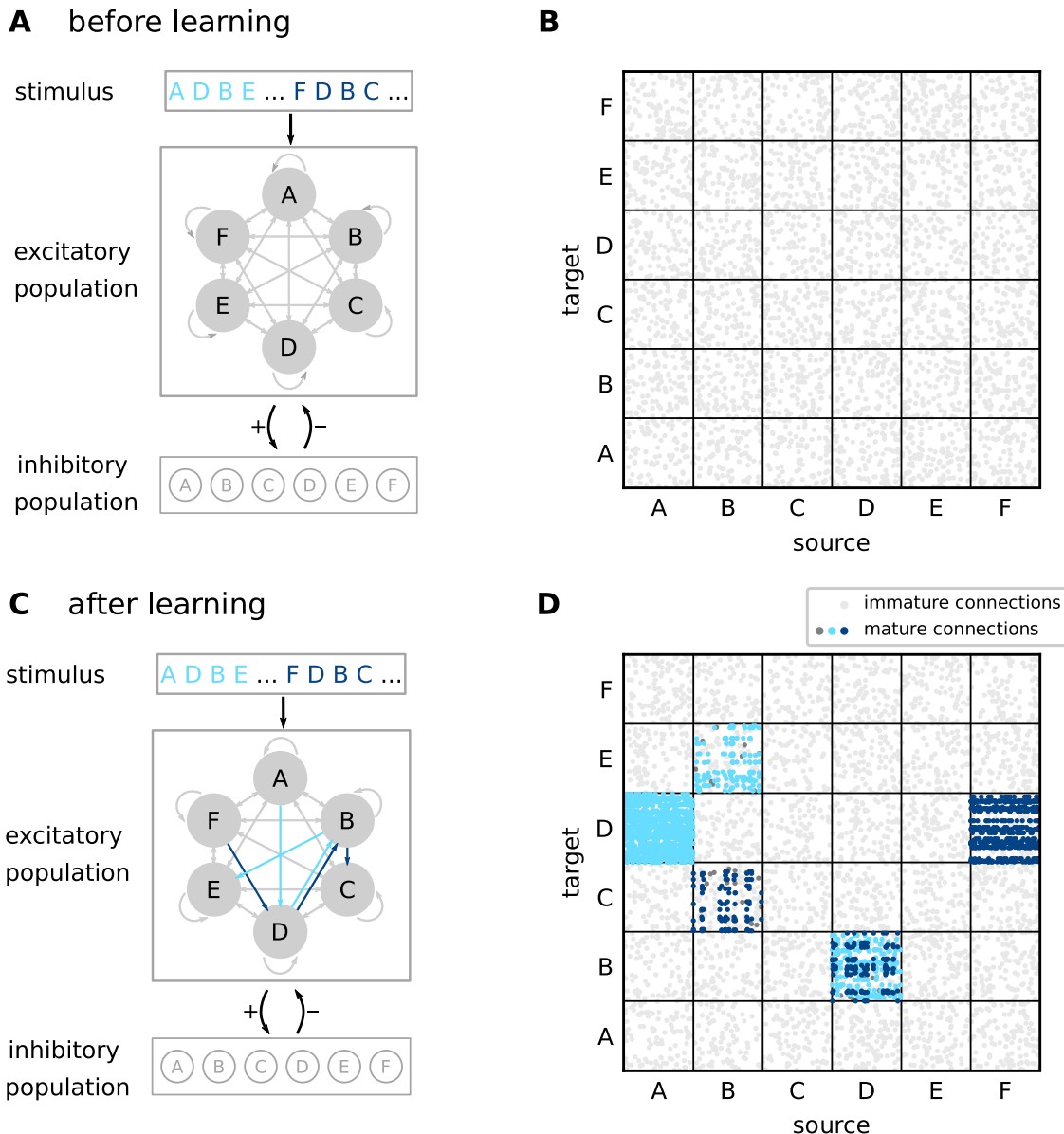

**Fig 2. Sketch of the network structure. A)** The architecture constitutes a recurrent network of excitatory and inhibitory neurons. Excitatory neurons are stimulated by external sources providing sequence-element specific inputs "A","D", etc. The excitatory neuron population is composed of subpopulations containing neurons with identical stimulus preference (gray circles). Connections between and within the excitatory subpopulations are random and sparse. Inhibitory neurons are mutually unconnected. Each neuron in the inhibitory population is recurrently connected to a specific subpopulation of excitatory neurons. **B)** Initial connectivity matrix for excitatory connections to excitatory neurons (EE connections). Target and source neurons are grouped into stimulus-specific subpopulations ("A",. . .,"F"). Before learning, the excitatory neurons are sparsely and randomly connected via immature synapses (light gray dots). **C)** During learning, sequence specific, sparsely connected subnetworks with mature synapses are formed (light blue arrows: {A, D, B, E}, dark blue arrows: {F, D, B, C}). **D)** EE connectivity matrix after learning. During the learning process, subsets of connections between subpopulations corresponding to subsequent sequence elements become mature and effective (light and dark blue dots). Mature connections are context specific (see distinct connectivity between subpopulations "D" and "B" corresponding to different sequences), thereby providing the backbone for a reliable propagation of sequence-specific activity. In panels B and D, only 5% of sequence non-specific EE connections are shown for clarity. Dark gray dots in panel D correspond to mature connections between neurons that remain silent after learning. For details on the network structure, see Tables 1 and 2.

containing neurons with identical stimulus preference ("receptive field"; see below). Each subpopulation $\mathcal{M}_k$ thereby represents a specific element within a sequence (Fig 2A and 2B). In the original TM model [14], a single sequence element is represented by multiple ($L$) subpopulations ("minicolumns"). For simplicity, we identify the number $M$ of subpopulations with the number of elements required for a specific set of sequences, such that each sequence element is encoded by just one subpopulation ($L = 1$). All neurons within a subpopulation $\mathcal{M}_k$ are recurrently connected to a subpopulation-specific inhibitory neuron $k \in \mathcal{I}$. The inhibitory neurons in $\mathcal{I}$ are mutually unconnected. The subdivision of excitatory neurons into stimulus-specific subpopulations defines how external inputs are fed to the network (see next paragraph), but does not affect the potential excitatory connectivity, which is homogeneous and not subpopulation specific.

**External inputs.**   During the prediction mode, the network is driven by an ensemble $\mathcal{X} = \{x_1, \ldots, x_M\}$ of $M$ external inputs, representing inputs from other brain areas, such as thalamic sources or other cortical areas. Each of these external inputs $x_k$ represents a specific sequence element ("A", "B", ...), and feeds all neurons in the subpopulation $\mathcal{M}_k$ with the corresponding stimulus preference. The occurrence of a specific sequence element $\zeta_{i,j}$ at time $t_{i,j}$ is modeled by a single spike $x_k(t) = \delta(t - t_{i,j})$ generated by the corresponding external source $x_k$. Subsequent sequence elements $\zeta_{i,j}$ and $\zeta_{i,j+1}$ within a sequence $s_i$ are presented with an inter-stimulus interval $\Delta T = t_{i,j+1} - t_{i,j}$. Subsequent sequences $s_i$ and $s_{i+1}$ are separated in time by an inter-sequence time interval $\Delta T_{\text{seq}} = t_{i+1,1} - t_{i,C_i}$. During the replay mode, we present only a cue signal encoding for first sequence elements $\zeta_{i,1}$ at times $t_{i,1}$. Subsequent cues are separated in time with an inter-cue time interval $\Delta T_{\text{cue}} = t_{i+1,1} - t_{i,1}$. In the absence of any other (inhibitory) inputs, each external input spike is strong enough to evoke an immediate response spike in all target neurons $i \in \mathcal{M}_k$. Sparse activation of the subpopulations in response to the external inputs is achieved by a winner-take-all mechanism implemented in the form of inhibitory feedback (see Sequence learning and prediction).

**Neuron and synapse model.**   In the original TM model [14], excitatory (pyramidal) neurons are described as abstract three-state systems that can assume an active, a predictive, or a non-active state. State updates are performed in discrete time. The current state is fully determined by the external input in the current time step and the network state in the previous step. Each TM neuron is equipped with a number of dendrites (segments), modeled as coincidence detectors. The dendrites are grouped into distal and proximal dendrites. Distal dendrites receive inputs from other neurons in the local network, whereas proximal dendrites are activated by external sources. Inputs to proximal dendrites have a large effect on the soma and trigger the generation of action potentials. Individual synaptic inputs to a distal dendrite, in contrast, have no direct effect on the soma. If the total synaptic input to a distal dendritic branch at a given time step is sufficiently large, the neuron becomes predictive. This dynamic mimics the generation of dendritic action potentials (dAPs), NMDA spikes [18–20]), which result in a long-lasting depolarization (∼50–500 ms) of the somata of neocortical pyramidal neurons.

In contrast to the original study, the model proposed here employs neurons with continuous-time dynamics. For all types of neurons, the temporal evolution of the membrane potential is given by the leaky integrate-and-fire model Eq (10). The total synaptic input current of excitatory neurons is composed of currents in distal dendritic branches, inhibitory currents, and currents from external sources. Inhibitory neurons receive only inputs from excitatory neurons in the same subpopulation. Individual spikes arriving at dendritic branches evoke alpha-shaped postsynaptic currents, see Eq (12). The dendritic current includes an additional nonlinearity describing the generation of dAPs: if the dendritic current $I_{\text{ED}}$ exceeds a threshold

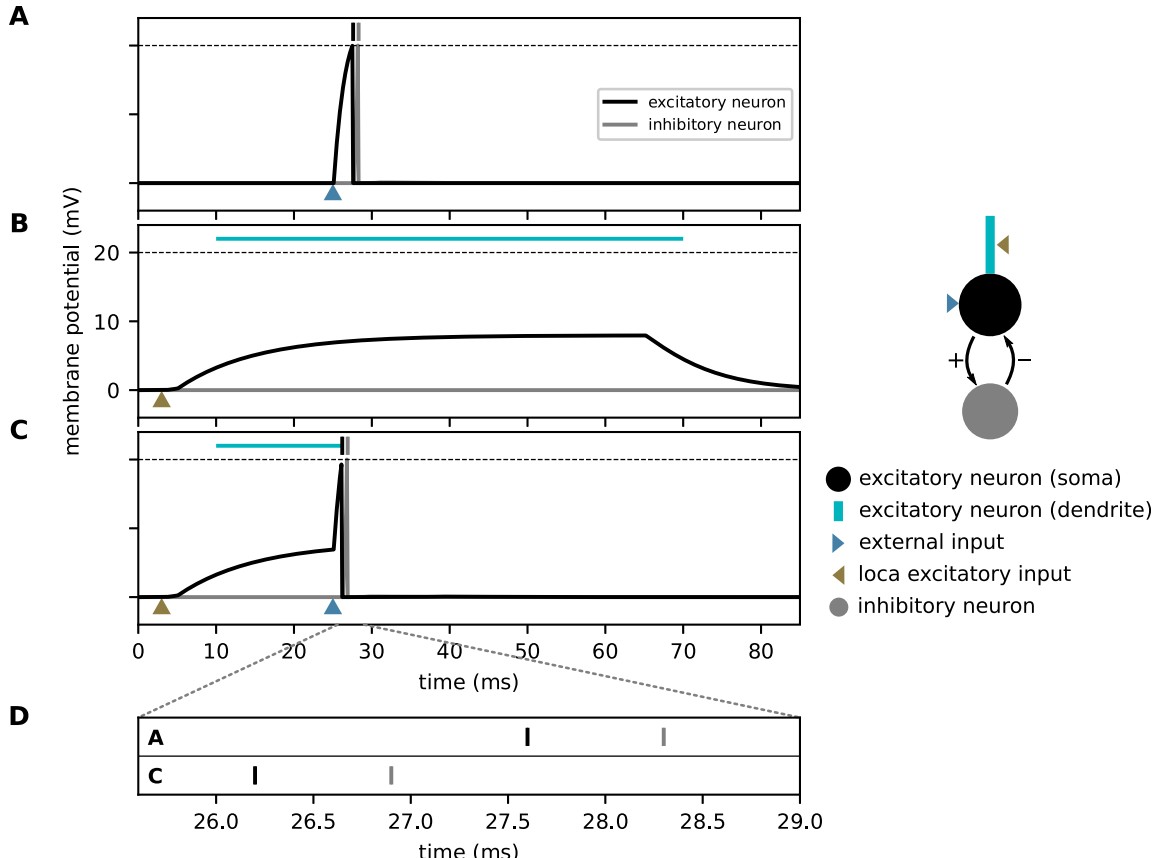

**Fig 3. Effect of dendritic action potentials (dAP) on the firing response to an external stimulus.** Membrane-potential responses to an external input (blue arrow, **A**), a strong dendritic input (brown arrow, **B**) triggering a dAP, and a combination of both (**C**). Black and gray vertical bars mark times of excitatory and inhibitory spikes, respectively. The horizontal dashed line marks the spike threshold $\theta_E$. The horizontal light blue lines depict the dAP plateau. **D)** Magnified view of spike times from panels A and C. A dAP preceding the external input (as in panel C) can speed up somatic, and hence, inhibitory firing, provided the time interval between the dAP and the external input is in the right range. The excitatory neuron is connected bidirectionally to an inhibitory neuron (see sketch on the right).

$\theta_{dAP}$, it is instantly set to a the dAP plateau current $I_{dAP}$, and clamped to this value for a period of duration $\tau_{dAP}$, see Eq (16). This plateau current leads to a long lasting depolarization of the soma (see Fig 3B). The dAP threshold $\theta_{dAP}$ is chosen such that the co-activation of $\gamma$ neurons with mature connections to the target neuron reliably triggers a dAP. In this work, we use a single dendritic branch per neuron. However, the model could easily be extended to include multiple dendritic branches. External and inhibitory inputs to excitatory neurons as well as excitatory inputs to inhibitory neurons trigger exponential postsynaptic currents, see Eqs (13)–(15). Similar to the original implementation, an external input strongly depolarizes the neurons and causes them to fire. To this end, the external weights $J_{EX}$ are chosen to be supra-threshold (see Fig 3A). Inhibitory interactions implement the WTA described in Sequence learning and prediction. The weights $J_{IE}$ of excitatory synapses on inhibitory neurons are chosen such that the collective firing of a subset of $\rho$ excitatory neurons in the corresponding sub-population causes the inhibitory neuron to fire. The weights $J_{EI}$ of inhibitory synapses on excitatory neurons are strong such that each inhibitory spike prevents all excitatory neurons in the same subpopulation that have not generated a spike yet from firing. All synaptic time constants, delays and weights are connection-type specific (see Table 1).

**Plasticity dynamics.** Both in the original [14] and in our model, the lateral excitatory connectivity between excitatory neurons (EE connectivity) is dynamic and shaped by a Hebbian structural plasticity mechanism mimicking principles known from the neuroscience literature [21–25]. All other connections are static. The dynamics of the EE connectivity is determined by the time evolution of the permanences $P_{ij}$ $(i, j \in \mathcal{E})$, representing the synapse maturity, and the synaptic weights $J_{ij}$. Unless the permanence $P_{ij}$ exceeds a threshold $\theta_P$, the synapse $\{j \rightarrow i\}$ is immature, with zero synaptic weight $J_{ij} = 0$. Upon threshold crossing, $P_{ij} \geq \theta_P$, the synapse becomes mature, and its weight is assigned a fixed value $J_{ij} = W$ $(\forall i, j)$. Overall, the permanences evolve according to a Hebbian plasticity rule: the synapse $\{j \rightarrow i\}$ is potentiated, i.e., $P_{ij}$ is increased, if the activation of the postsynaptic cell $i$ is immediately preceded by an activation of the presynaptic cell $j$. Otherwise, the synapse is depressed, i.e., $P_{ij}$ is decreased. At the beginning of the learning process or during relearning, the activity in the individual subpopulations is non-sparse. Hebbian learning alone would therefore lead to the strengthening of all existing synapses between two subsequently activated subpopulations, irrespective of the context these two subpopulations participate in. After learning, the subsets of neurons that are activated by a sequence element recurring in different sequences would therefore largely overlap. As a consequence, it becomes harder to distinguish between different contexts (histories) based on the activation patterns of these subsets. The original TM model [14] avoids this loss of context sensitivity by restricting synaptic potentiation to a small subset of synapses between a given pair of source and target subpopulations: if there are no predictive target neurons, the original algorithm selects a "matched" neuron from the set of active postsynaptic cells as the one being closest to becoming predictive, i.e., the neuron receiving the largest number of synaptic inputs on a given dendritic branch from the set of active presynaptic cells (provided this number is sufficiently large). Synapse potentiation is then restricted to this set of matched neurons. In case there are no immature synapses, the "least used" neuron or a randomly chosen neuron is selected as the "matched" cell, and connected to the winner cell of the previously active subpopulation. Restricting synaptic potentiation to synapses targeting such a subset of "matched" neurons is difficult to reconcile with biology. It is known that inhibitory inputs targeting the dendrites of pyramidal cells can locally suppress backpropagating action potentials and, hence, synaptic potentiation [26]. A selection mechanism based on such local inhibitory circuits would however involve extremely fast synapses and require fine-tuning of parameters. The model presented in this work circumvents the selection of "matched" neurons and replaces this with a homeostatic mechanism controlled by the postsynaptic dAP rate. In the following, the specifics of the plasticity dynamics used in this study are described in detail.

Within the interval $[P_{\text{min},ij}, P_{\text{max}}]$, the dimensionless permanences $P_{ij}(t)$ evolve according to a combination of an additive spike-timing-dependent plasticity (STDP) rule [27] and a homeostatic component [28, 29]:

$$
\begin{aligned}
P_{\text{max}}^{-1} \frac{dP_{ij}}{dt} &= \lambda_+ \sum_{\{t_i^*\}'} x_j(t)\delta(t - [t_i^* + d_{\text{EE}}])\, I(t_i^*, \Delta t_{\text{min}}, \Delta t_{\text{max}}) \\
&\quad - \lambda_- \sum_{\{t_j^*\}} y_i \delta(t - t_j^*) \\
&\quad + \lambda_{\text{h}} \sum_{\{t_i^*\}'} (z^* - z_i(t))\delta(t - t_i^*)\, I(t_i^*, \Delta t_{\text{min}}, \Delta t_{\text{max}}).
\end{aligned} \tag{1}
$$

At the boundaries $P_{\text{min},ij}$ and $P_{\text{max}}$, $P_{ij}(t)$ is clipped. While the maximum permanences $P_{\text{max}}$ are identical for all EE connections, the minimal permanences $P_{\text{min},ij}$ are uniformly distributed in the interval $[P_{0,\text{min}}, P_{0,\text{max}}]$ to introduce a form of persistent heterogeneity. The first term on

the right-hand side of Eq (1) corresponds to the spike-timing dependent synaptic potentiation triggered by the postsynaptic spikes at times $t_i^* \in \{t_i^*\}'$. Here, $\{t_i^*\}' = \{t_i^* | \forall t_j^* : t_i^* - t_j^* + d_{EE} \geq \Delta t_{\min}\}$ denotes the set of all postsynaptic spike times $t_i^*$ for which the time lag $t_i^* - t_j^* + d_{EE}$ exceeds $\Delta t_{\min}$ for all presynaptic spikes $t_j^*$. The indicator function $I(t_i^*, \Delta t_{\min}, \Delta t_{\max})$ ensures that the potentiation (and the homeostasis; see below) is restricted to time lags $t_i^* - t_j^* + d_{EE}$ in the interval $(\Delta t_{\min}, \Delta t_{\max})$ to avoid a growth of synapses between synchronously active neurons belonging to the same subpopulation, and between neurons encoding for the first elements in different sequences; see Eq 17. Note that the potentiation update times lag the somatic postsynaptic spike times by the delay $d_{EE}$, which is here interpreted as a purely dendritic delay [27, 30]. The potentiation increment is determined by the dimensionless potentiation rate $\lambda_+$, and the spike trace $x_j(t)$ of the presynaptic neuron $j$, which is updated according to

$$\frac{dx_j}{dt} = -\tau_+^{-1} x_j(t) + \sum_{t_j^*} \delta(t - t_j^*). \tag{2}$$

The trace $x_j(t)$ is incremented by unity at each spike time $t_j^*$, followed by an exponential decay with time constant $\tau_+$. The potentiation increment $\Delta P_{ij}$ at time $t_i^*$ therefore depends on the temporal distance between the postsynaptic spike time $t_i^*$ and all presynaptic spike times $t_j^* \leq t_i^*$ (STDP with all-to-all spike pairing; [27]). The second term in Eq (1) represents synaptic depression, and is triggered by each presynaptic spike at times $t_j^* \in \{t_j^*\}$. The depression decrement $y_i = 1$ is treated as a constant, independently of the postsynaptic spike history. The depression magnitude is parameterized by the dimensionless depression rate $\lambda_-$. The third term in Eq (1) corresponds to a homeostatic control triggered by postsynaptic spikes at times $t_i^* \in \{t_i^*\}'$. Its overall impact is parameterized by the dimensionless homeostasis rate $\lambda_h$. The homeostatic control enhances or reduces the synapse growth depending on the dAP trace $z_i(t)$ of neuron $i$, the low-pass filtered dAP activity updated according to

$$\frac{dz_i}{dt} = -\tau_h^{-1} z_i(t) + \sum_k \delta(t - t_{dAP,i}^k). \tag{3}$$

Here, $\tau_h$ represents the homeostasis time constant, and $t_{dAP,i}^k$ the onset time of the $k$th dAP in neuron $i$. According to Eq (1), synapse growth is boosted if the dAP activity $z_i(t)$ is below a target dAP activity $z^*$. Conversely, high dAP activity exceeding $z^*$ reduces the synapse growth

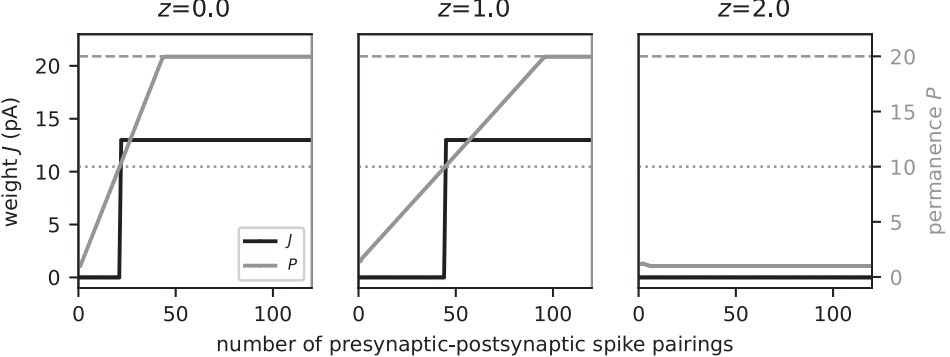

**Fig 4. Homeostatic regulation of the spike-timing-dependent structural plasticity by the dAP activity.** Evolution of the synaptic permanence (gray) and weight (black) during repetitive presynaptic-postsynaptic spike pairing for different levels of the dAP activity. In the depicted example, presynaptic spikes precede the postsynaptic spikes by 40 ms for each spike pairing. Consecutive spike pairs are separated by a 200 ms interval. In each panel, the postsynaptic dAP trace is clamped at a different value: $z = 0$ (left), $z = 1$ (middle), $z = 2$ (right). The dAP target activity is fixed at $z^* = 1$. The horizontal dashed and dotted lines mark the maximum permanence $P_{\max}$ and the maturity threshold $\theta_P$, respectively.

(Fig 4). This homeostatic regulation of the synaptic maturity controlled by the postsynaptic dAP activity constitutes a variation of previous models [28, 29] describing 'synaptic scaling' [31–33]. It counteracts excessive synapse formation during learning driven by Hebbian structural plasticity. In addition, the combination of Hebbian plasticity and synaptic scaling can introduce a competition between synapses [28, 29]. Here, we exploit this effect to ensure that synapses are generated in a context specific manner, and thereby reduce the overlap between neuronal subpopulations activated by the same sequence element occurring in different sequences. To this end, the homeostasis parameters $z^* = 1$ and $\tau_h$ are chosen such that each neuron tends to become predictive, i.e., generate a dAP, at most once during the presentation of a single sequence ensemble of total duration $((C - 1)\Delta T + \Delta T_{seq})S$ (see Network model). The time constant $\tau_h$ is hence adapted to the parameters of the task. For sequence sets I and II and the default inter-stimulus interval $\Delta T = 40$ ms, it is set to $\tau_h = 440$ ms and $\tau_h = 1560$ ms, respectively. In section Dependence of prediction performance on the sequence speed, we study the effect of the sequence speed (inter-stimulus interval $\Delta T$) on the prediction performance for a given network parameterization. For these experiments, $\tau_h = 440$ ms is therefore fixed even though the inter-stimulus interval $\Delta T$ is varied.

The prefactor $P_{max}^{-1}$ in Eq (1) ensures that all learning rates $\lambda_+$, $\lambda_-$ and $\lambda_h$ are measured in units of the maximum permanence $P_{max}$.

**Constraints on potential connectivity.** The sequence processing capabilities of the proposed network model rely on its ability to form sequence specific subnetworks based on the skeleton provided by the random potential connectivity. On the one hand, the potential connectivity must not be too diluted to ensure that a subset of neurons representing a given sequence element can establish sufficiently many mature connections to a second subset of neurons representing the subsequent element. On the other hand, a dense potential connectivity would promote overlap between subnetworks representing different sequences, and thereby slow down the formation of context specific subnetworks during learning (see Sequence learning and prediction). Here, we therefore identify the minimal potential connection probability $p$ guaranteeing the existence of network motifs with a sufficient degree of divergent-convergent connectivity.

Consider the subset $\mathcal{P}_{ij}$ of $\rho$ excitatory neurons representing the $j$th sequence element $\zeta_{ij}$ in sequence $s_i$ (see Task and training protocol and Network model). During the learning process, the plasticity dynamics needs to establish mature connections from $\mathcal{P}_{ij}$ to a second subset $\mathcal{P}_{i,j+1}$ of neurons in another subpopulation representing the subsequent element $\zeta_{i,j+1}$. Each neuron in $\mathcal{P}_{i,j+1}$ must receive at least $c = \lceil \theta_{dAP}/W \rceil$ inputs from $\mathcal{P}_{ij}$ to ensure that synchronous firing of the neurons in $\mathcal{P}_{ij}$ can evoke a dAP after synapse maturing. For a random, homogeneous potential connectivity with connection probability $p$, the probability of finding these $c$ potential connections for some arbitrary target neuron is given by

$$q(c; \rho, p) = \sum_{k=c}^{\rho} \binom{\rho}{k} p^k (1-p)^{\rho-k}. \tag{4}$$

For a successful formation of sequence specific subnetworks during learning, the sparse subset $\mathcal{P}_{ij}$ of presynaptic neurons needs to recruit at least $\rho$ targets in the set of $n_E$ neurons representing the subsequent sequence element (Fig 5A). The probability of observing such a divergent-convergent connectivity motif is given by

$$u(\rho; c, p, n_E) = \sum_{l=\rho}^{n_E} \binom{n_E}{l} q^l (1-q)^{n_E-l}. \tag{5}$$

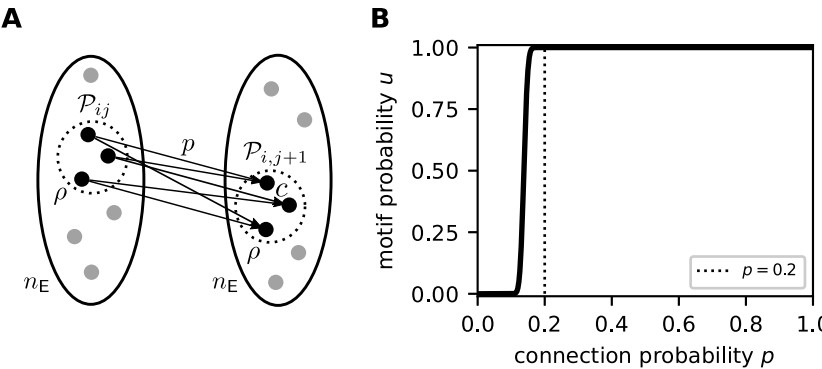

**Fig 5. Existence of divergent-convergent connectivity motifs in a random network. A)** Sketch of the divergent-convergent potential connectivity motif required for the formation of sequence specific subnetworks during learning. See main text for details. **B)** Dependence of the motif probability $u$ on the connection probability $p$ for $n_E = 150$, $c = 5$, and $\rho = 20$ (see Table 2). The dotted vertical line marks the potential connection probability $p = 0.2$ used in this study.

Note that the above described motif does not require the size of the postsynaptic subset $\mathcal{P}_{i,j+1}$ to be exactly $\rho$. Eq (5) constrains the parameters $p$, $c$, $n_E$ and $\rho$ to ensure such motifs exist in a random network. Fig 5B illustrates the dependence of the motif probability $u$ on the connection probability $p$ for our choice of parameters $n_E$, $c$, and $\rho$. For $p \geq 0.2$, the existence of the divergent-convergent connectivity motif is almost certain ($u \approx 1$). For smaller connection probabilities $p < 0.2$, the motif probability quickly vanishes. Hence, $p = 0.2$ constitutes a reasonable choice for the potential connection probability.

**Network realizations and initial conditions.** For every network realization, the potential connectivity and the initial permanences are drawn randomly and independently. All other parameters are identical for different network realizations. The initial values of all state variables are given in Tables 1 and 2.

**Simulation details.** The network simulations are performed in the neural simulator NEST [34] under version 3.0 [35]. The differential equations and state transitions defining the excitatory neuron dynamics are expressed in the domain specific language NESTML [36, 37] which generates the required C++ code for the dynamic loading into NEST. Network states are synchronously updated using exact integration of the system dynamics on a discrete-time grid with step size $\Delta t$ [38]. The full source code for the implementation with a list of other software requirements is available at Zenodo: https://doi.org/10.5281/zenodo.5578212.

## Task performance measures

To assess the network performance, we monitor the dendritic currents reporting predictions (dAPs) as well as the somatic spike times of excitatory neurons. To quantify the prediction error, we identify for each last element $\zeta_{i,C_i}$ in a sequence $s_i$ all excitatory neurons that have generated a dAP in the time interval $(t_{i,C_i} - \Delta T, t_{i,C_i})$, where $t_{i,C_i}$ and $\Delta T$ denote the time of the external input corresponding to the last sequence element $\zeta_{i,C_i}$ and the inter-stimulus interval, respectively (see Task and training protocol and Network model). All subpopulations $\mathcal{M}_k$ with at least $\rho/2$ neurons generating a dAP are considered "predictive". The prediction state of the network is encoded in an $M$ dimensional binary vector $\boldsymbol{o}$, where $o_k = 1$ if the $k$th subpopulation is predictive, and $o_k = 0$ else. The

$$\text{prediction error} = \frac{1}{L}\sqrt{\sum_{k=1}^{M}(o_k - v_k)^2} \tag{6}$$

is defined as the Euclidean distance between $o$ and the binary target vector $v$ representing the pattern of external inputs for each last element $\zeta_{i,C_i}$, normalized by the number $L$ of subpopulations per sequence element. Furthermore, we assess the

$$\text{false positive rate} = \frac{1}{L}\sum_{k=1}^{M}\Theta(o_k - v_k) \tag{7}$$

and the

$$\text{false negative rate} = \frac{1}{L}\sum_{k=1}^{M}\Theta(v_k - o_k), \tag{8}$$

where $\Theta(\cdot)$ denotes the Heaviside function. In addition to these performance measures, we monitor for each last sequence element the level of sparsity by measuring the ratio between the number of active neurons and the total number $Ln_E$ of neurons representing this element. During learning, we expose the network repetitively to the same set $\{s_1, \ldots, s_S\}$ of sequences for a number of training episodes $K$. To obtain the total prediction performance in each episode, we average the prediction error, the false negative and false positive rates, as well as the level of sparsity across the set of sequences.

## Results

### Sequence learning and prediction

According to the Temporal Memory (TM) model, sequences are stored in the form of specific paths through the network. Prediction and replay of sequences correspond to a sequential sparse activation of small groups of neurons along these paths. Non-anticipated stimuli are signaled in the form of non-sparse firing of these groups. This subsection describes how the model components introduced in Network model interact and give rise to the network structure and behavior postulated by TM. For illustration, we here consider a simple set of two partly overlapping sequences {A, D, B, E} and {F, D, B, C} corresponding to the sequence set I (see Fig 1B).

The initial sparse, random and immature network connectivity (Fig 2A and 2B) constitutes the skeleton on which the sequence-specific paths will be carved out during the learning process. To guarantee a successful learning, this initial skeleton must be neither too sparse nor too dense (see Methods). Before learning, the presentation of a particular sequence element causes all neurons with the corresponding stimulus preference to reliably and synchronously fire a somatic action potential due to the strong, suprathreshold external stimulus (Fig 3A). All other subpopulations remain silent (see Fig 6A and 6B). The lateral connectivity between excitatory neurons belonging to the different subpopulations is subject to a form of Hebbian structural plasticity. Repetitive and consistent sequential presentation of sequence elements turns immature connections between successively activated subpopulations into mature connections, and hence leads to the formation of sequence-specific subnetworks (see Fig 2C and 2D). Synaptic depression prunes connections not supporting the learned pattern, thereby reducing the chance of predicting wrong sequence items (false positives).

During the learning process, the number of mature connections grows to a point where the activation of a certain subpopulation by an external input generates dendritic action potentials (dAPs), a "prediction", in a subset of neurons in the subsequent subpopulation (blue neurons in Fig 6C). The dAPs generate a long-lasting depolarization of the soma (Fig 3B). When receiving an external input, these depolarized neurons fire slightly earlier as compared to non-depolarized (non-predictive) neurons (Fig 3A, 3B and 3D). If the number of predictive neurons

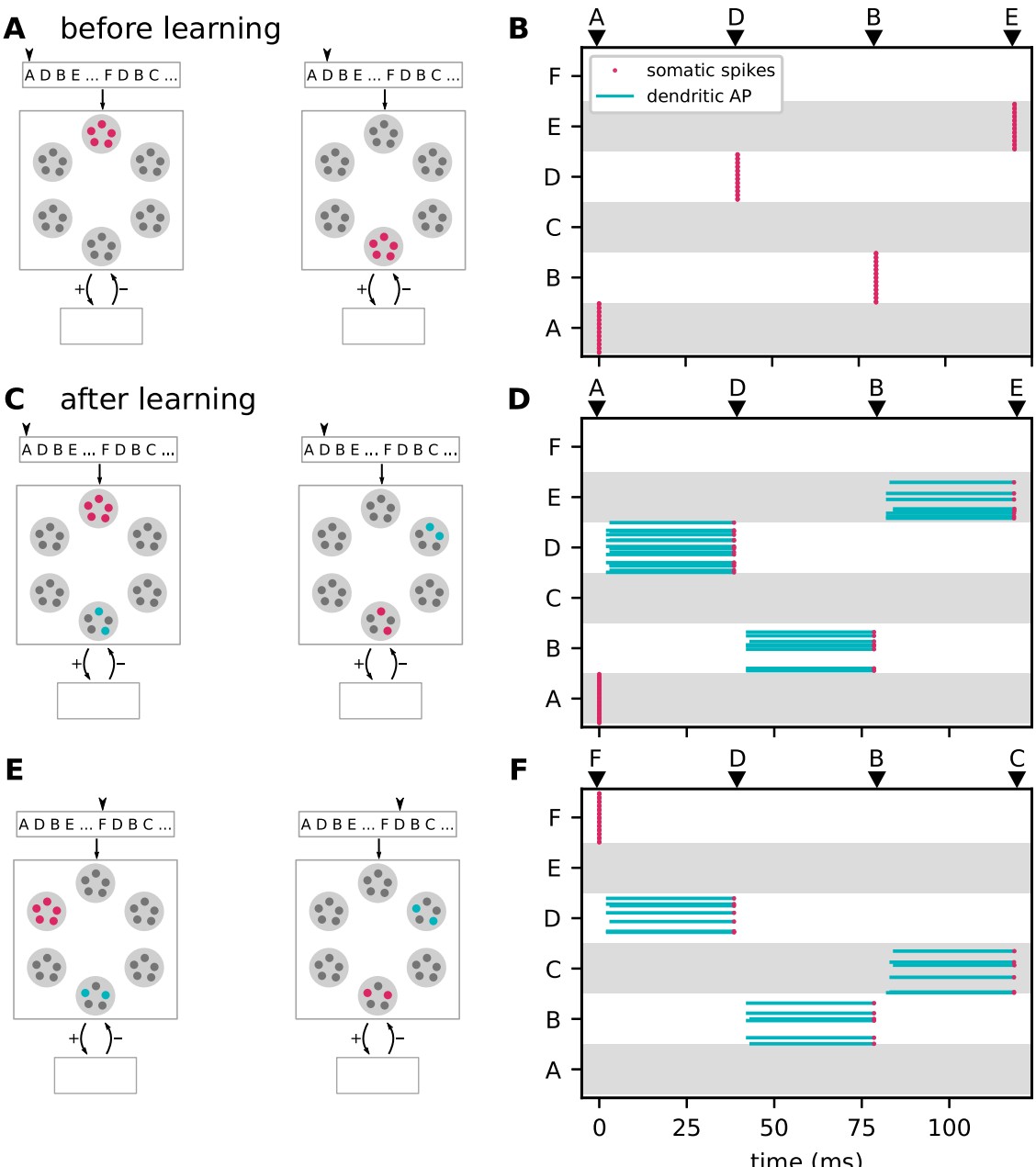

**Fig 6. Context specific predictions.** Sketches (left column) and raster plots of network activity (right column) before (top row) and after learning of the two sequences {A, D, B, E} and {F, D, B, C} (middle and bottom rows). In the left column, large light gray circles depict the excitatory subpopulations (same arrangement as in Fig 2). Red, blue and gray circles mark active, predictive and silent neurons, respectively. In the right column, red dots and blue lines mark somatic spikes and dAP plateaus, respectively. Type and timing of presented stimuli are depicted by black arrows. **A,B)** Snapshots of network activity upon subsequent presentation of the sequence elements "A" and "D" (panel A), and network activity in response to presentation of the entire sequence {A, D, B, E} (panel B) before learning. All neurons in the stimulated subpopulations become active. **C,D)** Same as panels A and B, but after learning. Presenting the first element "A" causes all neurons in the corresponding subpopulations to fire. Activation of these neurons triggers dAPs (predictions) in a subset of neurons representing the subsequent element "D". When the next element "D" is presented, only these predictive neurons become active, leading to predictions in the subpopulation representing the subsequent subpopulation ("B"), etc. **E,F)** Same as panels C and D, but for sequence {F, D, B, C}. The subsets of neurons representing "D" and "B" activated during sequences {A, D, B, E} and {F, D, B, C} are distinct, i.e., context specific. For clarity, panels B, D, and F show only a fraction of excitatory neurons (30%).

within a subpopulation is sufficiently large, their advanced spikes (Fig 3C) initiate a fast and strong inhibitory feedback to the entire subpopulation, and thereby suppress subsequent firing of non-predictive neurons in this population (Fig 6C and 6D). Owing to this winner-take-all dynamics, the network generates sparse spiking in response to predicted stimuli, i.e., if the external input coincides with a dAP-triggered somatic depolarization. In the presence of a non-anticipated, non-predicted stimulus, the neurons in the corresponding subpopulation fire collectively in a non-sparse manner, thereby signaling a "mismatch".

In the model presented in this study, the initial synapse maturity levels, the permanences, are randomly chosen within certain bounds. During learning, connections with a higher initial permanence mature first. This heterogeneity in the initial permanences enables the generation of sequence specific sparse connectivity patterns between subsequently activated neuronal sub-populations (Fig 2D). For each pair of sequence elements in a given sequence ensemble, there is a unique set of postsynaptic neurons generating dAPs (Fig 6D). These different activation patterns capture the context specificity of predictions. When exposing a network that has learned the two sequences {A, D, B, E} and {F, D, B, C} to the elements "A" and "F", different subsets of neurons are activated in "D" and "B". By virtue of these sequence specific activation patterns, stimulation by {A, D, B} or {F, D, B} leads to correct predictions "E" or "C", respectively (Fig 6C–6F).

Heterogeneity in the permanences alone, however, is not sufficient to guarantee context specificity. The subsets of neurons activated in different contexts may still exhibit a considerable overlap. This overlap is promoted by Hebbian plasticity in the face of the initial non-sparse activity, which leads to a strengthening of connections to neurons in the postsynaptic population in an unspecific manner (Fig 7A and 7B). Moreover, the reoccurrence of the same sequence elements in different co-learned sequences initially causes higher firing rates of the neurons in the respective populations ("D" and "B" in Fig 7). As a result, the formation of unspecific connections would even be accelerated if synapse formation was driven by Hebbian plasticity alone. The model in this study counteracts this loss of context specificity by supplementing the plasticity dynamics with a homeostatic component, which regulates synapse growth based on the rate of postsynaptic dAPs. This form of homeostasis prevents the same neuron from becoming predictive multiple times within the same set of sequences, and thereby reduces the overlap between subsets of neurons activated within different contexts (Fig 7C and S4 Fig). To further aid the formation of context specific paths, the density of the initial potential connectivity skeleton is set close to the minimum value ensuring the existence of the connectivity motifs required for a faithful prediction (see Methods).

## Prediction performance

To quantify the sequence prediction performance, we repetitively stimulate the network with the sequences in sequence set I (see Task and training protocol), and continuously monitor the prediction error, the false-positive and false-negative rates, as well as the fraction of active stimulated neurons as a measure of encoding sparsity (Fig 8; Task performance measures). To ensure the performance results are not specific to a single network, the evaluation is repeated for a number of randomly instantiated network realizations with different initial potential connectivities. At the beginning of the learning process, all neurons of a stimulated subpopulation collectively fire in response to the external input. Non-stimulated neurons remain silent. As the connectivity is still immature at this point, no dAPs are triggered in postsynaptic neurons, and, hence, no predictions are generated. As a consequence, the prediction error, the false-negative rate and the number of active neurons (in stimulated populations) are at their maximum, and the false positive rate is zero (Fig 8). During the first training episodes, the

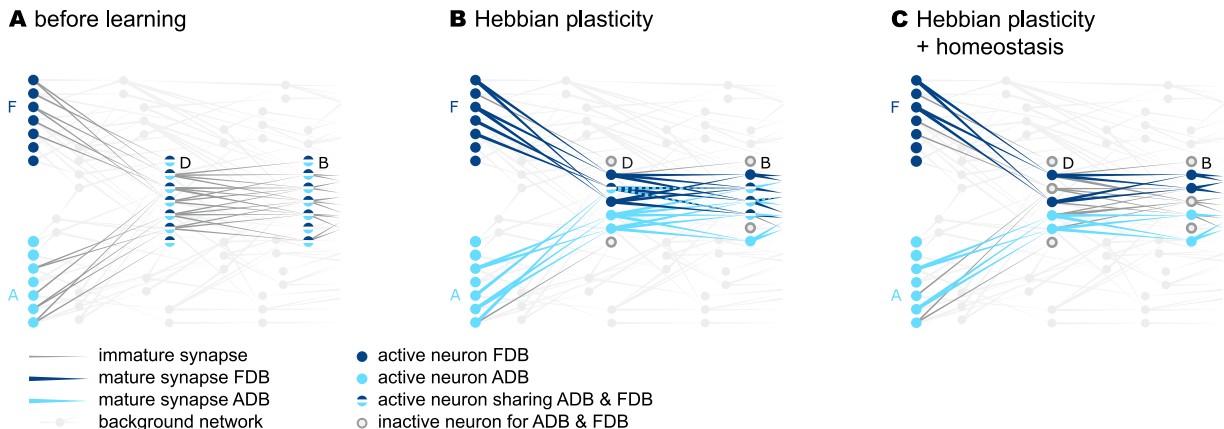

**Fig 7. dAP-rate homeostasis enhances context specificity. A)** Sketch of subpopulations of excitatory neurons representing the elements of the two sequences {F, D, B} and {A, D, B}, depicted by light and dark blue colors, respectively. Before learning, the connections between the subpopulations are immature (gray lines). Hence, for each element presentation, all neurons in the respective subpopulations fire (filled circles). **B)** Hebbian plasticity drives the formation of mature connections between subpopulations representing successive sequence elements (colored lines), and leads to sparse firing. The sets of neurons contributing to the two sequences partly overlap. **C)** Incorporating dAP-rate homeostasis reduces this overlap in the activation patterns.

consistent collective firing of subsequently activated populations leads to the formation of mature connections as a result of the Hebbian structural plasticity. Upon reaching of a critical number of mature synapse, first dAPs (predictions) are generated in postsynaptic cells (in Fig 8, this happens after about 10 learning episodes). As a consequence, the false negative rate decreases, and the stimulus responses become more sparse. At this early phase of the learning, the predictions of upcoming sequence elements are not yet context specific (for sequence set I, non-sparse activity in "B" triggers a prediction in both "E" and "C", irrespective of the context). Hence, the false-positive rate transiently increases. As the context specific connectivity is not consolidated at this point, more and more presynaptic subpopulations fail at triggering dAPs in their postsynaptic targets when they switch to sparse firing. Therefore, the false-positive rate decreases again, and the false-negative rate increases. In other words, there exists a negative feedback loop in the interim learning dynamics where the generation of predictions

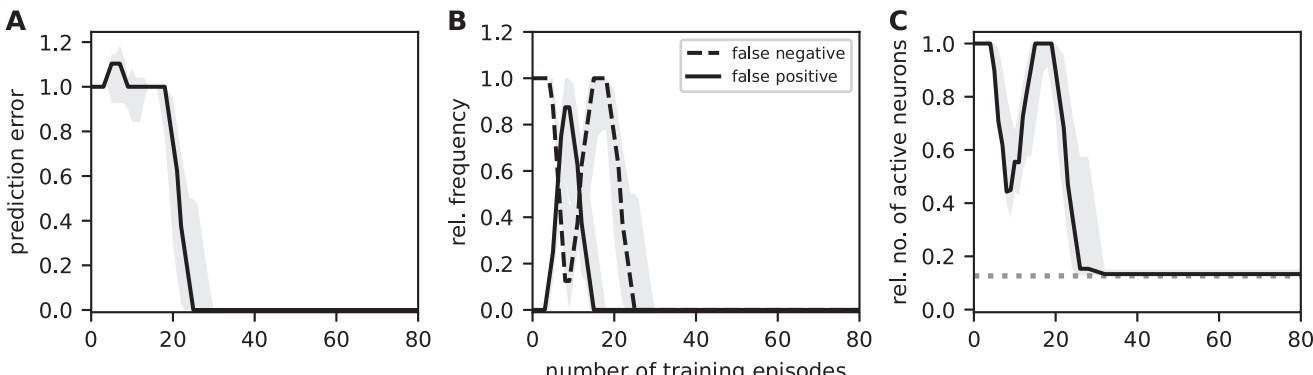

**Fig 8. Sequence prediction performance for sequence set I.** Dependence of the sequence prediction error (**A**), the false-positive and false-negative rates (**B**), and the number of active neurons relative to the subpopulation size (**C**) on the number of training episodes during repetitive stimulation with sequence set I (see Task and training protocol). Curves and error bands indicate the median as well as the 5% and 95% percentiles across an ensemble of 5 different network realizations, respectively. All prediction performance measures are calculated as a moving average over the last 4 training episodes. The dashed gray horizontal line in panel C depicts the target sparsity level $\rho/(Ln_E)$. Inter-stimulus interval $\Delta T = 40$ ms. See Table 2 for remaining parameters.

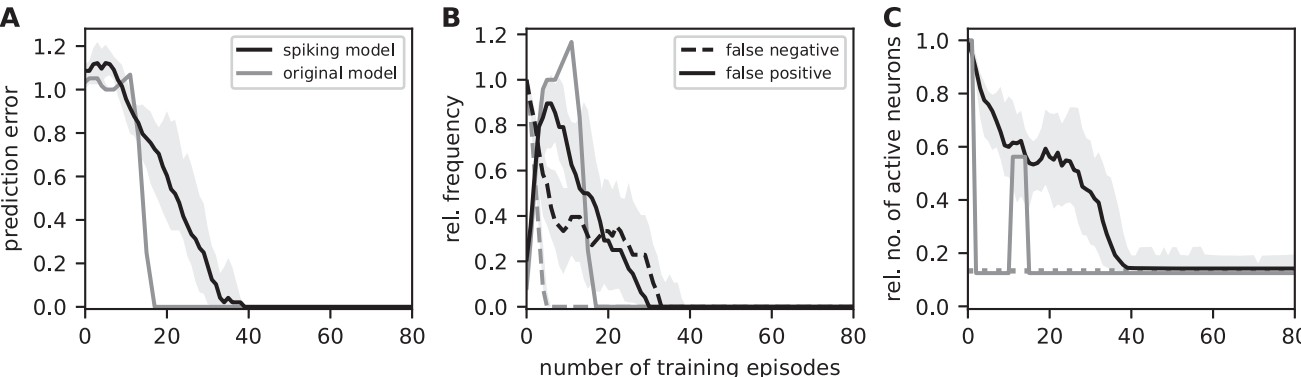

**Fig 9. Sequence prediction performance for sequence set II and comparison with original model.** Same figure arrangement, training and measurement protocol as in Fig 8. Data obtained during repetitive stimulation of the network with sequence set II (see Task and training protocol). Gray curves depict results obtained using the original (non-spiking) TM model from [14] with adapted parameters (see S1 Table). The dashed gray horizontal line in panel C depicts the target sparsity level $\rho/(Ln_{\mathrm{E}})$.

leads to an increase in sparsity which, in turn, causes prediction failures (and, hence, non-sparse firing). With an increasing number of training episodes, synaptic depression and homeostatic regulation increase context selectivity and thereby break this loop. Eventually, sparse firing of presynaptic populations is sufficient to reliably trigger predictions in their post-synaptic targets. For sequence set I, the total prediction error becomes zero and the stimulus responses maximally sparse after about 30 training episodes (Fig 8). For a time resolved visualization of the learning dynamics, see S1 Video.

Up to this point, we illustrated the model's sequence learning dynamics and performance for a simple set of two sequences (sequence set I). In the following, we assess the network's sequence prediction performance for a more complex sequence set (II) composed of five high-order sequences (see Task and training protocol), each consisting of five elements. This sequence set is comparable to the one used in [14], but contains a larger amount of overlap between sequences. The overall pattern of the learning dynamics resembles the one reported for sequence set I (Fig 9). The prediction error, the false-positive and false-negative rates as well as the sparsity measure vary more smoothly, and eventually converge at minimal levels after about 40 training episodes. To compare the spiking TM model with the original, non-spiking TM model, we repeat the experiment based on the simulation code provided in [14], see S1 Table. With our parameterization, the learning rates $\lambda_{+}$ and $\lambda_{-}$ of the spiking model are by a factor of about 10 smaller than in the original model. As a consequence, learning sequence set II with the original model converges faster than with the spiking model (compare black and gray curves in Fig 9). The ratio in learning speeds, however, is not larger than about 2. Increasing the learning rates, i.e., the permanence increments, would speed up the learning process in the spiking model, but bears the risk that a large fraction of connections mature simultaneously. This would effectively overwrite the permanence heterogeneity which is essential to form context specific connectivity patterns (see Sequence learning and prediction). As a result, the network performance would decrease. The original model avoids this problem by limiting the number of potentiated synapses in each update step (see "Plasticity dynamics" in Network model).

In sequence sets I and II, the maximum sequence order is 2 and 3, respectively. For the two sequences {E, N, D, I, J} and {L, N, D, I, K} in sequence set II, for example, predicting element "J" after activation of "I" requires remembering the element "E", which occured three steps back into the past. The TM model can cope with sequences of much higher order. Each

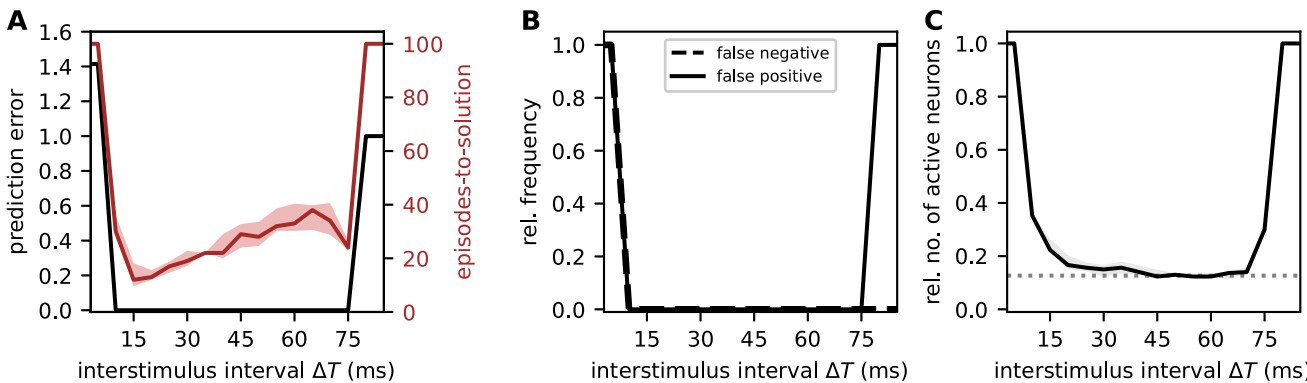

**Fig 10. Effect of sequence speed on network performance.** Dependence of the sequence prediction error, the learning speed (episodes-to-solution; **A**), the false-positive and false-negative rates (**B**), and the number of active neurons relative to the subpopulation size (**C**) on the inter-stimulus interval $\Delta T$ after 100 training episodes. Curves and error bands indicate the median as well as the 5% and 95% percentiles across an ensemble of 5 different network realizations, respectively. Same task and network as in Fig 8.

sequence element in a particular context activates a specific pattern, i.e., a specific subset of neurons. The number of such patterns that can be learned is determined by the size of each subpopulation and the sparsity [39]. In a sequence with repeating elements, such as {ABBBBBC}, the maximum order is limited by this number. Without repeating elements, the order could be arbitrarily high provided the number of subpopulations matches or exceeds the number of distinct characters. In S3 Fig, we demonstrate successful learning of two sequences {A, D, B, G, H, I, J, K, L, M, N, E}, {F, D, B, G, H, I, J, K, L, M, N, C} of order 10.

## Dependence of prediction performance on the sequence speed

The reformulation of the original TM model in terms of continuous-time dynamics allows us to ask questions related to timing aspects. Here, we investigate the sequence processing speed by identifying the range of inter-stimulus intervals $\Delta T$ that permit a successful prediction performance (Fig 10). The timing of the external inputs affects the dynamics of the network in two respects. First, reliable predictions of sequence elements can only be made if the time interval $\Delta T$ between two consecutive stimulus presentations is such that the second input coincides with the somatic depolarization caused by the dAP triggered by the first stimulus. Second, the formation of sequence specific connections by means of the spike-timing-dependent structural plasticity dynamics depends on $\Delta T$.

If the external input does not coincide with the somatic dAP depolarization, i.e., if $\Delta T$ is too small or to large, the respective target population responds in a non-sparse, non-selective manner (mismatch signal; Fig 10C), and in turn, generates false positives (Fig 10B). For small $\Delta T$, the external stimulus arrives before the dAP onset, i.e., before it is predicted. In consequence, the false negative rate is high. For large $\Delta T$, the false negative rate remains low as the network is still generating predictions (Fig 10B). The inter-stimulus interval $\Delta T$ in addition affects the formation of sequence specific connections due to the dependence of the plasticity dynamics on the timing of pre- and postsynaptic spikes, see Eqs (1) and (2). Larger $\Delta T$ results in smaller permanence increments, and thereby a slow-down of the learning process (red curve in Fig 10A).

Taken together, the model predicts a range of optimal inter-stimulus interval $\Delta T$ (Fig 10A). For our choice of network parameters, this range spans intervals between 10 ms and 75 ms. The lower bound depends primarily on the synaptic time constant $\tau_{EE}$, the spike transmission

delay $d_{EE}$, and the membrane time constant $\tau_m$. The upper bound is mainly determined by the dAP plateau duration $\tau_{dAP}$.

## Sequence replay

So far, we studied the network in the predictive mode, where the network is driven by external inputs and generates predictions of upcoming sequence elements. Another essential component of sequence processing is sequence replay, i.e., the autonomous generation of sequences in response to a cue signal (see Task and training protocol). After successful learning, the network model presented in this study is easily configured into the replay mode by increasing the neuronal excitability, such that the somatic depolarization caused by a dAP alone makes the neuron fire a somatic spike. Here, this is implemented by lowering the somatic spike threshold $\theta_E$ of the excitatory neurons. In the biological system, this increase in excitability could, for example, be caused by the effect of neuromodulators [40, 41], additional excitatory inputs from other brain regions implementing a top-down control, e.g, attention [42, 43], or propagating waves during sleep [44, 45].

The presentation of the first sequence element activates dAPs in the subpopulation corresponding to the expected next element in a previously learned sequence. Due to the reduced firing threshold in the replay mode, the somatic depolarization caused by these dAPs is sufficient to trigger somatic spikes during the rising phase of this depolarization. These spikes, in turn, activate the subsequent element. This process repeats, such that the network autonomously reactivates all sequence elements in the correct order, with the same context specificity and sparsity level as in the predictive mode (see Fig 11A and 11B). The latency between the activation of subsequent sequence elements is determined by the spike transmission delay $d_{EE}$, the synaptic time constant $\tau_{EE}$, the membrane time constant $\tau_{m,E}$, the synaptic weights $J_{EE,ij}$, the dAP current plateau amplitude $I_{dAP}$, and the somatic firing threshold $\theta_E$. For sequences that can be successfully learned (see previous section), the time required for replaying the entire sequence is independent of the inter-stimulus interval $\Delta T$ employed during learning (Fig 11C).

As shown in the previous section, sequences cannot be learned if the inter-stimulus interval $\Delta T$ is too small or too large. For small $\Delta T$, connections between subpopulations corresponding

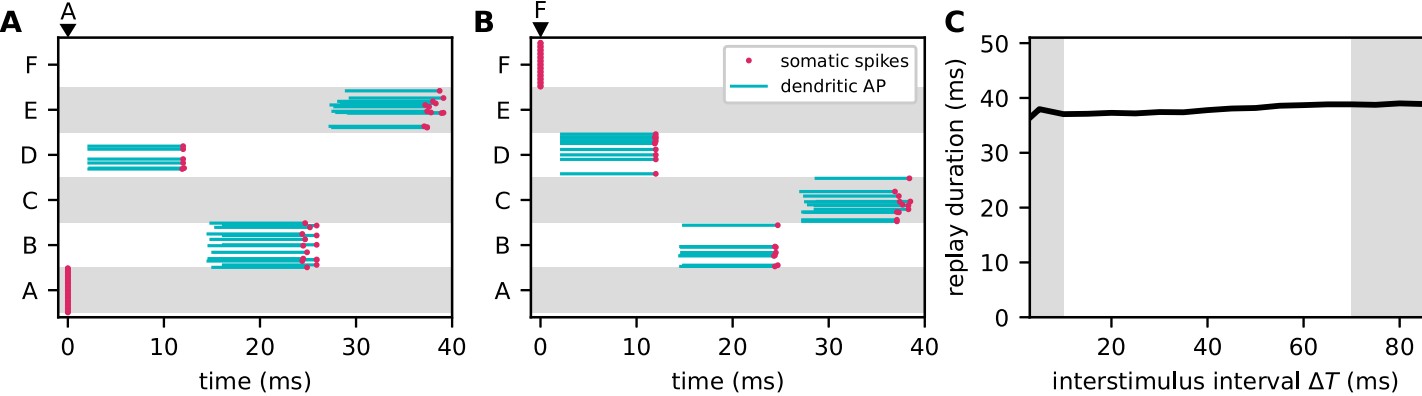

**Fig 11. Sequence replay dynamics and speed.** Autonomous replay of the sequences {A, D, B, E} (**A**) and {F, D, B, C} (**B**), initiated by stimulating the subpopulations "A" and "F", respectively. Red dots and blue lines mark somatic spikes and dAP plateaus, respectively, for a fraction of neurons (30%) within each subpopulation. During learning, the inter-stimulus interval $\Delta T$ is set to 40 ms. **C)** Dependence of the sequence replay duration on the inter-stimulus interval $\Delta T$ during learning. Replay duration is measured as the difference between the mean firing times of the populations representing the first and last elements in a given sequence. Gray areas mark regions with low prediction performance (see Dependence of prediction performance on the sequence speed). Error bands represent the mean ± standard deviation of the prediction error across 5 different network realizations. Same network and training set as in Fig 8.

to subsequent elements are strongly potentiated by the Hebbian plasticity due to the consistent firing of pre- and postsynaptic populations during the learning process. The network responses are, however, non-sparse, as the winner-take-all mechanism cannot be invoked during the learning (Fig 10C). In the replay mode, sequences are therefore replayed in a non-sparse and non-context specific manner (left gray region in Fig 11C). Similarly, connections between subsequent populations are slowly potentiated for very large $\Delta T$. With sufficiently long learning, sequences can still be replayed in the right order, but the activity is non-sparse and therefore not context specific (right gray region in Fig 11C).

## Discussion

### Summary

In this work, we reformulate the Temporal Memory (TM) model [14] in terms of biophysical principles and parameters. We replace the original discrete-time neuronal and synaptic dynamics with continuous-time models with biologically interpretable parameters such as membrane and synaptic time constants and synaptic weights. We further substitute the original plasticity algorithm with a more biologically plausible mechanism, relying on a form of Hebbian structural plasticity, homeostatic control, and sparse random connectivity. Moreover, our model implements a winner-take-all dynamics based on lateral inhibition that is compatible with the continuous-time neuron and synapse models. We show that the revised TM model supports successful learning and processing of high-order sequences with a performance similar to the one of the original model [14].

A new aspect that we investigated in the context of our work is sequence replay. After learning, the model is able to replay sequences in response to a cue signal. The duration of sequence replay is independent of the sequence speed during training, and determined by the intrinsic parameters of the network. In general, sequence replay is faster than the sequence presentation during learning, consistent with sequence compression and fast replay observed in hippocampus [46–48] and neocortex [6, 49].

Finally, we identified the range of possible sequence speeds that guarantee a successful learning and prediction. Our model predicts an optimal range of processing speeds (inter-stimulus intervals) with lower and upper bounds constrained by neuronal and synaptic parameters (e.g., firing threshold, neuronal and synaptic time constants, coupling strengths, potentiation time constants). Within this range, the number of required training episodes is proportional to the inter-stimulus interval $\Delta T$.

### Relationship to other models

The model presented in this work constitutes a recurrent, randomly connected network of neurons with predefined stimulus preferences. The model learns sequences in an unsupervised manner using local learning rules. This is in essence similar to several other spiking neuronal network models for sequence learning [9–12, 50]. The new components employed in this work are dendritic action potentials (dAPs) and Hebbian structural plasticity. We use structural plasticity to be as close as possible to the original model, and Hebbian forms of this are also known from the literature [21, 22, 25]. However, preliminary results show that classical (non-structural) spike-timing-dependent plasticity (STDP) can yield similar performance (see S1 Fig). Dendritic action potentials are instrumental for our model for two reasons. First, they effectively lower the threshold for coincidence detection and thereby permit a reliable and robust propagation of sparse activity [51, 52]. In essence, our model bears similarities to the classical synfire chain [53], one difference being that our mature network is not a simple feedforward network but has an abundance of recurrent connections. As shown in [54], a stable

propagation of synchronous activity requires a minimal number of neurons in each synfire group. Without active dendrites, this minimal number is in the range of ~100 for plausible single-cell and synaptic parameters. In our (and in the original TM) model, coincidence detection happens in the dendrites. The number of presynaptic spikes needed to trigger a dAP is small, of the order of 10 [55–57]. This helps to reduce redundancy (only a small number of neurons needs to become active) and to increase the capacity of the network (the number of different patterns that can be learned is increased with pattern sparsity; [39]). Second, dAPs equip neurons with a third type of state (next to the quiescent and the firing state): the predictive state, i.e., a long lasting (~ 50–200 ms) strong depolarization of the soma. Due to the prolonged depolarization of the soma, the inter-stimulus interval can be much larger than the synaptic time constants and delays. An additional benefit of dAPs, which is not exploited in the current version of our model, is that they equip individual neurons with more possible states if they comprise more than one dendritic branch. Each branch constitutes an independent pattern detector. The response of the soma may depend on the collective predictions in different dendritic branches. A single neuron could hence perform the types of computations that are usually assigned to multilayer perceptrons, i.e., small networks [58, 59].

Similar to a large class of other models in the literature, the TM network constitutes a recurrent network in the sense that the connectivity before and after learning forms loops at the subpopulation level. Recurrence in the immature connectivity permits the learning of arbitrary sequences without prior knowledge of the input data. In particular, recurrent connections enable the learning of sequences with repeating elements (such as in {A, B, B, C} or {A, B, C, B}). Further, bidirectional connections between subpopulations are needed to learn sequences where pairs of elements occur in different orders (such as in {A, B, C}, {D, C, B}). Apart from providing the capability to learn sequences with all possible combinations of sequence elements, recurrent connections play no further functional role in the current version of the TM model. They may, however, become more important for future versions of the model enabling the learning of sequence timing and duration (see below).

Most of the existing models have been developed to replay learned sequences in response to a cue signal. The TM model can perform this type of pattern completion, too. In addition, it can act as a quiet, sparsely active observer of the world that becomes highly active only in the case of unforeseen, non-anticipated events. In this work, we didn't directly analyze the network's mismatch detection performance. However, this could be easily achieved by equipping each population with a "mismatch" neuron that fires if a certain fraction of neurons in the population fires (threshold detectors). In our model, predicted stimuli result in sparse firing due to inhibitory feedback (WTA). For unpredicted stimuli, this feedback is not effective, resulting in non-sparse firing indicating a mismatch. In [60], a similar mechanism is employed to generate mismatch signals for novel stimuli. In this study, the strength of the inhibitory feedback needs to be learned by means of inhibitory synaptic plasticity. In our model, the WTA mechanism is controlled by the predictions (dAPs) and implemented by static inhibitory connections. Furthermore, the model in [60] can learn a set of elements, but not the order of these elements in the sequence.

In contrast to other sequence learning models [9, 11], our model is not able to learn an element specific timing and duration of sequence elements. The model in [9] relies on a clock network, which activates sequence elements in the correct order and with the correct timing. With this architecture, different sequences with different timings would require separate clock networks. Our model learns both sequence contents and order for a number of sequences without any auxiliary network. In an extension of our model, the timing of sequence element could be learned by additional plastic recurrent connections within each subpopulation. The model in [11] can learn and recall higher-order sequences with limited history by means of an

additional reservoir network with sparse readout. The TM model presents a more efficient way of learning and encoding the context in high-order sequences, without prior assignment of context specificity to individual neuron populations [9], and without additional network components (such as reservoir networks in [11]).

An important sequence processing component that is not addressed in our work is the capability of identifying recurring sequences within a long stream of inputs. In the literature, this process is referred to as chunking, and constitutes a form of feature segmentation [3]. Sequence chunking has been illustrated, for example, in [61, 62]. Similar to our model, the network model in [61, 62] is composed of neurons with dendritic and somatic compartments, with the dendritic activity signaling a prediction of somatic spiking. Recurrent connections in their model improve the context specificity of neuronal responses, and thereby permit a context dependent feature segmentation. The model can learn high order sequences, but the history is limited. Although not explicitly tested here, our model is likely to be able to perform chunking if sequences are presented randomly across trials and without breaks. If the order of sequences is not systematic across trials, connections between neurons representing different sequences are not strengthened by spike-timing-dependent potentiation. Consecutive sequences are therefore not merged and remain distinct.

An earlier spiking neural network version of the HTM model has already been devised in [63]. It constitutes a proof-of-concept study demonstrating that the HTM model can be ported to an analog-digital neuromorphic hardware system. It is restricted to small simplistic sequences and does not address the biological plausibility of the TM model. In particular, it does not offer a solution to the question of how the model can perform online learning by known biological ingredients. Our study delivers a solution for this based on local plasticity rules and permits a direct implementation on a neuromorphic hardware system.

## Limitations and outlook

The model developed in this study serves as a proof of concept demonstrating that the TM algorithm proposed in [14] can be implemented using biological ingredients. While it is still fairly simplistic, it may provide the basis for a number of future extensions.

Our results on the sequence processing speed revealed that the model presented here can process fast sequences with inter-stimulus intervals $\Delta T$ up to $\sim$75 ms. This range of processing speeds is relevant in many behavioral contexts such as motor generation, vision (saccades), music perception and generation, language, and many others [64]. However, slow sequences with inter-stimulus intervals beyond several hundreds of milliseconds cannot be learned by this model with biologically plausible parameters. This is problematic as behavioral time scales are often larger [64, 65]. By increasing the duration $\tau_{\mathrm{dAP}}$ of the dAP plateau, the upper bound for $\Delta T$ could be extended to 500 ms, and maybe beyond [66]. However, for such long intervals, the synaptic potentiation would be very slow, unless the time constant $\tau_+$ of the structural STDP is increased and the depression rate $\lambda_-$ is adapted accordingly. Furthermore, while our model explains the fast replay observed in the hippocampus and cortex, it is not able to learn an element specific timing and duration of sequence elements [5, 67, 68]. This could be overcome by equipping the model with a working memory mechanism, which maintains the activity of the subpopulations for behaviorally relevant time scales [9, 69].

In the current version of the model, the number of subpopulations, the number of neurons within each subpopulation, the number of dendritic branches per neuron, as well as the number of synapses per neuron are far from realistic [14]. The number of sequences that can be successfully learned in this network is hence rather small. In addition, the current work is focusing on sequence processing at a single abstraction level, not accounting for a hierarchical

network and task structure with both bottom-up and top-down projections. A further simplification in this work is that the lateral inhibition within a subpopulation is mediated by a single interneuron with unrealistically strong and fast connections to and from the pool of excitatory neurons. In future versions of this model, this interneuron could be replaced by a recurrently connected network of inhibitory neurons, thereby permitting more realistic weights, and simultaneously speeding up the interaction between inhibitory and excitatory cells by virtue of the fast-tracking property of such networks [70]. Similarly, the external inputs in our model are represented by single spikes, which are passed to the corresponding target population by a strong connection, and thereby lead to an immediate synchronous spike response. Replacing each external input by a population of synchronously firing neurons would be a more realistic scenario without affecting the model dynamics. The external neurons could even fire in a non-synchronous, rate modulated fashion, provided the spike responses of the target populations remain nearly synchronous and can coincide with the dAP-triggered somatic depolarization (see S6 Fig). The current version of the model relies on a nearly synchronous immediate response to ensure that a small set of ($\sim$ 20) active neurons can reliably trigger postsynaptic dAPs, and that the predictive neurons (those depolarized by the dAPs) consistently fire earlier as compared to the non-predictive neurons, such that they can be selected by the WTA dynamics. Non-synchronous responses could possibly lead to a reliable generation of dAPs in postsynaptic neurons, but would require large active neuron populations (loss of sparsity) or unrealistically strong synaptic weights. The temporal separation between predictive and non-predictive neurons becomes harder for non-synchronous spiking. In future versions of the model, it could potentially be achieved by increasing the dAP plateau potential, and simultaneously equipping the excitatory neurons with a larger membrane time constant, such that non-depolarized neurons need substantially longer to reach the spike threshold. Increasing the dAP plateau potential, however, makes the model more sensitive to background noise (see below). Note that, in our model, only the immediate initial spike response needs to be synchronous. After successfully triggering the WTA circuit, the winning neurons could –in principle– continue firing in an asynchronous manner (for example, due the working-memory dynamics mentioned above). Similarly, long lasting or tonic external inputs could lead to repetitive firing of the neurons in the TM network. As long as these repetitive responses remain nearly synchronous, the network performance is likely to be preserved.

In the predictive mode, the statistics of the spiking activity generated by our model is primarily determined by the temporal structure of the external inputs. Upon presentation of a sequence element, a specific subset of excitatory neurons fires a single volley of synchronous spikes. If the stimulus is predicted, this subset is small. The spike response is therefore highly sparse both in time and space, in line with experimental findings [71]. For simplicity and illustration, the sequences in this study are presented in a serial manner with fixed order, and fixed inter-sequence and inter-element (inter-stimulus) intervals. As a consequence, the single-neuron spike responses are highly regular. The in-vivo spiking activity in cortical networks, in contrast, exhibits a high degree of irregularity [72]. A more natural presentation of sequences with irregular order and timing trivially leads to more irregular spike responses in our model. As long as the inter-stimulus intervals fall into the range depicted in Fig 10, the model can learn and predict irregular sequences. Spiking activity in the cortex is not only irregular, but also fairly asynchronous in the sense that the average level of synchrony for randomly chosen pairs of neurons is low [73, 74]. This, however, is not necessarily the case for any subset of neurons and at any point in time. It is well known that cortical neurons can systematically synchronize their firing with millisecond precision in relation to behaviorally relevant events (see, e.g., [75]). As demonstrated in [76], synchronous firing of small subsets of neurons may easily go unnoticed in the presence of subsampling. The model proposed in this study relies on

(near) synchronous firing of small subsets of neurons. In cases where the model processes large sets of sequences in parallel, this synchrony will hardly be observable if only a small fraction of neurons is monitored (see S5 Fig). After learning, different sequences are represented by distinct subnetworks with little overlap. Hence, the network can process multiple sequences at the same time with little interference between subnetworks. The model could even learn multiple sequences in parallel, provided there is no systematic across-trial dependency between the sequences presented simultaneously. We dedicate the task of testing these ideas to future studies. While the synchrony predicted by the TM model may hardly be observable in experimental data suffering from strong subsampling, the predicted patterns of spikes could be identifiable using methods accounting for both spatial and temporal dependencies in the spike data [76–78]. There are other factors that may contribute to a more natural spiking activity in extended versions of the model. First, equipping the model with a working memory mechanism enabling the learning of slow sequences and sequence timing (see above) would likely lead to sustained asynchronous irregular firing. Second, replacing the inhibitory neurons by recurrent networks of inhibitory neurons (see above) would generate asynchronous irregular activity in the populations of inhibitory neurons and thereby contribute variability in the spike responses of the excitatory neurons. Third, the model proposed here may constitute a module embedded into a larger architecture and receive irregular inputs from other components. As shown in the supplementary S6 and S7 Figs, the spiking activity and the prediction performance of the TM model are robust with respect to low levels of synaptic background activity, and, hence, membrane potential fluctuations reminiscent of those observed in vivo [79]. For an increasing level of noise, the learning speed decreases. For high noise levels leading to additional, non-task related background spikes, the dAP triggered plateau depolarization is overwritten, such that the WTA dynamics fails at selecting predictive neurons, ultimately leading to a loss of context specificity in the responses. Hence, the prediction performance degrades for large noise amplitudes. A potential application of introducing background noise is to allow the network to perform probabilistic computations [80], such as replaying sequences in the presence of ambiguous cues.

Similar to the original TM model, the response of the population representing the first element in a sequence is non-sparse, indicating that the first sequence element is not anticipated and can therefore not be predicted. If a given first sequence element reoccurs within the same sequence (say, "A" in {A, B, A, C}) or in other sequences (e.g., in {D, E, A, F}), the non-sparse response of the respective population to a first sequence element leads to a simultaneous prediction of all possible subsequent elements, i.e., the generation of false positives. These false predictions would lead to a pruning of functional synapses as a response of the homeostatic regulation to the increased dAP activity. This could be overcome by three possible mechanisms: a) synaptic normalization avoiding excessive synapse growth [81, 82], b) removing breaks between sequences, or c) sparse, sequence specific firing of subpopulations representing first elements. Results of applying the last mechanism are shown in S2 Fig, where dAPs are externally activated in random subsets of neurons in the populations representing first elements. In a more realistic hierarchical network, a similar effect could be achieved by top-down projections from a higher level predicting sequences of sequences.

In the original model, synapses targeting silent postsynaptic cells are depressed, even if the presynaptic neuron is inactive. This pruning process, the freeing of unused synaptic resources, increases the network capacity while ensuring context sensitivity. According to the structural plasticity dynamics employed in our study, synapse depression is bound to presynaptic spiking, similar to other implementations of (non-structural) STDP [30]. As a consequence, strong connections originating from silent presynaptic neurons are not depressed (dark gray dots in

Fig 2D). This may complicate or slow down the learning of new sequences, and could be overcome by synaptic normalization.

For the dAP-rate homeostasis used in this study, the target dAP rate is set to one to make sure that each neuron contributes at most one dAP during each training episode. As a consequence, the time constant of the dAP-rate homeostasis is adapted to the duration of a training episode, which is in the range of few seconds in this work. We are not aware of any biological mechanism that could account for such an adaptation. dAP-rate homeostasis is mediated by the intracellular calcium concentration, which, in turn, controls the synthesis of synaptic receptors, and hence, the synaptic strength. It is therefore known to be rather slow, acting on timescales of many minutes, hours or days [32, 33]. It is unclear to what extent the use of long homeostatic time constants and increased dAP target rates would alter the model performance. Alternatively, the dAP-rate homeostasis could be replaced by other mechanisms such as synaptic normalization.

## Conclusion

Our work demonstrates that the principle mechanisms underlying sequence learning, prediction, and replay in the TM model can be implemented using biologically plausible ingredients. By strengthening the link to biology, our implementation permits a more direct evaluation of the TM model predictions based on electrophysiological and behavioral data. Furthermore, this implementation allows for a direct mapping of the TM model on neuromorphic hardware systems.

## Supporting information

**S1 Table. Adapted parameters of the original TM model used for Fig 9.** Parameter names match those used in the original simulation code (https://github.com/numenta/htmpapers/tree/master/frontiers/why_neurons_have_thousands_of_synapses). Gray parameter names are those used in the spiking TM model.
(EPS)

**S1 Fig. Sequence prediction performance in the presence of conventional (non-structural) spike-timing dependent plasticity (STDP).** Dependence of the sequence prediction error (**A**), the false-positive and false-negative rates (**B**), and the number of active neurons relative to the subpopulation size (**C**) on the number of training episodes for sequence set II. Curves and error bands indicate the median as well as the 5% and 95% percentiles across an ensemble of 5 different network realizations, respectively. All prediction performance measures are calculated as a moving average over the last 4 training episodes. In this experiment, structural STDP is replaced by conventional STDP, i.e., the permanences $P_{ij}(t)$ and $P_{max}$ in Eq (1) are replaced by the synaptic weights $J_{EE,ij}(t)$ and $J_{max}$. The weights $J_{EE,ij}$ are restricted to the interval $[J_{min,ij}, J_{max}]$, and clipped at the boundaries. The minimal weights $J_{min,ij}$ are randomly and independently drawn from a uniform distribution between $J_{0,min}$ and $J_{0,max}$. The performance characteristics shown here are comparable to those obtained with structural STDP (see Fig 9 in Prediction performance). Parameters: $\Delta T = 40$ ms, $\lambda_+ = 0.43$, $\lambda_- = 0.0058$, $\lambda_h = 0.03$, $J_{0,min} = 0$pA, $J_{0,max} = 2$pA, $J_{max} = 12.98$pA. See Table 2 for remaining parameters.
(EPS)

**S2 Fig. Prediction performance for a sequence set with recurring first items.** Dependence of the sequence prediction error (**A**), the false positive frequency, the false negative frequency (**B**), and the number of active neurons relative to the subpopulation size (**C**) on the number of training episodes for a set of sequences $s_1 = \{B, D, I, C, H\}$, $s_2 = \{E, D, I, C, F\}$, $s_3 = \{F, B, C, A,$

H}, $s_4$ = {G, B, C, A, D}, $s_5$ = {E, C, I, H, A}, $s_6$ = {D, C, I, H, G} with recurring first items. Curves and error bands indicate the median as well as the 5% and 95% percentiles across 5 different network realizations, respectively. As a solution to the issue discussed in Limitations and outlook concerning the recurring of first sequence elements in other sequences or within the same sequence, the dAPs are externally activated in a random subset of neurons in the populations representing first elements. Inter-stimulus interval $\Delta T$ = 40 ms. All prediction performance measures are calculated as a moving average over the last 4 training episodes. Parameters: $\Delta T$ = 40 ms, $\lambda_+$ = 0.39, $\lambda_-$ = 0.0057, $\lambda_h$ = 0.034. See Table 2 for remaining parameters. (EPS)

**S3 Fig. Prediction performance for a sequence set with 10 overlapping elements.** Dependence of the sequence prediction error (**A**), the false positive frequency, the false negative frequency (**B**), and the number of active neurons relative to the subpopulation size (**C**) on the number of training episodes for a set of two sequences $s_1$ = {A, D, B, G, H, I, J, K, L, M, N, E} and $s_2$ = {F, D, B, G, H, I, J, K, L, M, N, C}. Curves and error bands indicate the median as well as the 5% and 95% percentiles across 5 different network realizations, respectively. Inter-stimulus interval $\Delta T$ = 40 ms. All prediction performance measures are calculated as a moving average over the last 4 training episodes. The parameters of the plasticity are identical to those reported in Table 2 for sequence set I. (EPS)

**S4 Fig. Effect of the dAP-rate homeostasis on the prediction performance.** Dependence of the prediction error (**A**) and the overlap in the activation pattern between the neurons representing the sequence element "G" in the context of sequences {A, D, B, G, H, E} and {F, D, B, G, H, C} (**B**) on the number of training episodes explored for two values of the homeostasis rate ($\lambda_h$). Curves and error bands indicate the median as well as the 5% and 95% percentiles across 5 different network realizations, respectively. Disabling the homeostasis control ($\lambda_h$ = 0.0) increases the overlap in the "G" activation pattern, which leads to a lost of context specificity and hence an increase in the prediction error (see Sequence learning and prediction). The parameters of the plasticity are identical to those reported in Table 2 for sequence set I. (EPS)

**S5 Fig. Asynchronous irregular firing in a (hypothetical) network processing multiple sequences in parallel. A**: Artificial spike data mimicking activity of a TM network processing $S$ = 10 sequences in parallel. Each sequence (right y-axis) is processed by a distinct subnetwork of 200 neurons, each composed of $C$ = 10 subpopulations. The horizontal gray lines separate the different subnetworks. Upon activation of a sequence element, $\rho$ = 20 neurons in the corresponding subpopulation synchronously fire a spike. Individual sequences are activated independently with rate 1 s$^{-1}$ at random times (Poisson point process with 200 ms deadtime). Inter-element intervals $\Delta T \sim \mathcal{U}(10\,\text{ms}, 80\,\text{ms})$ are randomly drawn from a uniform distribution (cf. Fig 10). The inset depicts a magnified view of a single activation of sequence 2. **B**: Same data as in A after random permutation of neuron identities.**C**: Spiking activity of a random subset of 100 neurons depicted in panel B. **D–F**: Distributions of single-neuron firing rates (D), inter-spike-interval variation coefficients (E; ISI CV), and spike-count correlation coefficients (F; binsize 10 ms) obtained from subsampled data shown in panel C for a total simulation time of 100 s (mean rate = 1 spikes/s, mean ISI CV = 0.8, mean correlation = 0.01). The data and analysis results shown here demonstrate that i) irregular sequence activation translates into irregular spiking, and ii) subsampling and the absence of prior knowledge of the network structure hide synchrony (but note the tiny peak at 1.0 in the distribution of

correlation coefficients). The combination of both effects hence leads to asynchronous irregular firing, reminiscent of in-vivo cortical activity.
(EPS)

**S6 Fig. Effects of background noise and non-synchronous stimulation on network activity. A–F)** Spiking activity before (panels A–C; 1st learning episode) and after learning sequence set I (panels D–F; 600th learning episode) in response to a presentation of sequence {A, D, B, E} without background noise (left) and in the presence of moderate (middle) or high synaptic background noise (right). External inputs are presented in the form of dispersed volleys of 50 spikes (black vertical bars at the top). Each of these spikes triggers an exponential synaptic input current in the target neurons with amplitude 134 pA and time constant 1 ms. Spike times in each spike volley are randomly drawn from a Gaussian distribution (width 4 ms), centered on the stimulus time. In each trial, all stimulated neurons receive the same realization of the Gaussian spike packet. Red dots and blue horizontal lines mark somatic spikes and dAPs, respectively. For clarity, only a fraction of 50% of excitatory neurons and external spikes are shown. Background noise to each excitatory neuron is provided in the form of balanced excitatory and inhibitory synaptic inputs, generated by distinct uncorrelated Poissonian spike sources (total rate per source $\nu = 10000$ s$^{-1}$). Background synapses are modeled as exponential postsynaptic currents (time constant $\tau_B = 1$ ms) with amplitudes $J = 0$ pA (left), 60 pA (middle), and 170 pA (right) for excitatory inputs, and $-J$ for inhibitory inputs, respectively. The mean background input $\mu = \tau_B \nu (J - J) = 0$ to each neuron vanishes due to the asymmetry in excitatory and inhibitory synaptic weights. The variance $\sigma^2 = \tau_B \nu J^2$ of the synaptic background current is modulated by adjusting the synaptic weight $J$ (left: $\sigma = 0$ pA, middle: $\sigma = 189$ pA, right: $\sigma = 537$ pA). **G,H,I)** Membrane potential traces of two neurons in the excitatory subpopulation "B" during the same time interval depicted in panels D–E for three noise levels $\sigma = 0$ pA (G), 189 pA (H), and 537 pA (I). One of the selected neurons (blue) is participating in the sequence, i.e, it generates a dAP and a somatic spike in response to sequence elements "D" and "B". The other neuron (orange) is not part of the sequence. The horizontal dashed lines and blue stars mark the threshold $\theta_E$ and the times of somatic spikes, respectively. Parameters: $\Delta T = 40$ ms, $\lambda_+ = 0.05$, $\lambda_- = 0.001$, $\lambda_h = 0.01$, $W = 23.6$ pA, $\Delta t_{min} = 20$ ms, $\tau_{dAP} = 40$ ms, $\tau_{ref,I} = 20$ ms, $J_{EI} = -9686.62$pA. See Table 2 for remaining parameters.
(EPS)

**S7 Fig. Effects of background noise and non-synchronous stimulation on prediction performance and sparsity for sequence set I.** Dependence of the sequence prediction error (**A**), the false positive and false negative rate (**B**), and the sparsity (number of active neurons relative to the subpopulation size, **C**) on the number of training episodes for three different noise amplitudes $\sigma = 0$ pA (black), 189 pA (blue), and 537 pA (gray). See caption of S6 Fig for details on the implementation of external inputs and background noise. Curves and error bands indicate the median as well as the 5% and 95% percentiles across 5 different network realizations, respectively. All prediction performance measures are calculated as a moving average over the last 4 training episodes. Same parameters as in S6 Fig.
(EPS)

**S1 Algorithm. Algorithmic description of the plasticity model, based on the algorithm proposed in [30].**
(EPS)

**S1 Video. Time resolved visualization of the learning dynamics.** Network activity (top) and connectivity (bottom) of the network during one learning episode. Each frame corresponds to a new training episode. In each learning episode, each of the two sequences {A, D, B, E} and

{F, D, B, C} is presented once (black arrows in the top panel). **Top panel**: Red dots and blue bars mark spike and dAP times for each neuron. Neurons are sorted according to stimulus preference (vertical axis). **Bottom panel**: Network connectivity before learning (left) and during the current training episode (right). Light gray and black dots represent immature and mature connections, respectively, for each pair of source and target neurons (sorted according to stimulus preference; see Sequence learning and prediction).
(MP4)

## Acknowledgments

The authors thank Rainer Waser for valuable discussions on the project, Charl Linssen for help with the neuron model implementation using NESTML, Abigail Morrison for suggestions on the plasticity dynamics, Danylo Ulianych and Dennis Terhorst for code review, and Sebastian B.C. Lehmann for assistance with graphic design. All network simulations were carried out with NEST (http://www.nest-simulator.org).

## Author Contributions

**Conceptualization:** Younes Bouhadjar, Dirk J. Wouters, Markus Diesmann, Tom Tetzlaff.

**Data curation:** Younes Bouhadjar.

**Formal analysis:** Younes Bouhadjar, Tom Tetzlaff.

**Funding acquisition:** Dirk J. Wouters, Markus Diesmann, Tom Tetzlaff.

**Investigation:** Younes Bouhadjar, Dirk J. Wouters, Markus Diesmann, Tom Tetzlaff.

**Methodology:** Younes Bouhadjar, Dirk J. Wouters, Markus Diesmann, Tom Tetzlaff.

**Project administration:** Younes Bouhadjar.

**Software:** Younes Bouhadjar.

**Supervision:** Dirk J. Wouters, Markus Diesmann, Tom Tetzlaff.

**Visualization:** Younes Bouhadjar.

**Writing – original draft:** Younes Bouhadjar, Dirk J. Wouters, Markus Diesmann, Tom Tetzlaff.

**Writing – review & editing:** Younes Bouhadjar, Dirk J. Wouters, Markus Diesmann, Tom Tetzlaff.

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
