## [Decision Letter · Decision Letter 0]

20 Dec 2021

Dear Mr Bouhadjar,

Thank you very much for submitting your manuscript "Sequence learning, prediction, and replay in networks of spiking neurons" for consideration at PLOS Computational Biology.

As with all papers reviewed by the journal, your manuscript was reviewed by members of the editorial board and by several independent reviewers. In light of the reviews (below this email), we would like to invite the resubmission of a significantly-revised version that takes into account the reviewers' comments.

We cannot make any decision about publication until we have seen the revised manuscript and your response to the reviewers' comments. Your revised manuscript is also likely to be sent to reviewers for further evaluation.

Sincerely,

Stefan Kiebel

Associate Editor

PLOS Computational Biology

Samuel Gershman

Deputy Editor

PLOS Computational Biology

Reviewer's Responses to Questions

**Comments to the Authors:**

Reviewer #1: The review is uploaded as an attachment

Reviewer #2: The paper from Bouhadjar et al. is asking an interesting question. The authors wonder how to combine several ‘universal’ tasks in the same network in a more biological implementation. In previous studies, these tasks were either studied separately, or in a somewhat artificial set-up. In their framework the homeostasis in particular, enhancing context specificity, has great merit. I would like to ask the authors, before any final recommendation is made, to answer some questions I had when reading the manuscript. In general the manuscript is of good quality. I also am happy that the authors are very honest about the limitations of the model, such as the limited replay capability.

Some major questions:

While combining several tasks is a feature of the model, previous work focusing on individual tasks may have the benefit that they can zoom in and mechanistically understand things in great detail. It seems that some part of the solution or implementation in these other studies could be shared with the solution in this paper. In this regard, you already cited resemblance with amongst others Cone and Shouval, eLife 2021, the sequence learning aspect. A more detailed discussion may be beneficial for the reader. As the authors are also interested in mismatch/prediction it would be good to discuss resemblance in this domain, for example: Schulz et al., eLife 2021. Finally I am wondering if there is any link to feature segmentation, as studied by for example the group of Tomoki Fukai (example paper is Asabuki and Fukai, Nat Comms 2020).

Is there a limit to the history/context memory? For example would ABCDEFG and HBCDEFJ give different predictions for the final element? In other words, would the homeostasis still be able to separate the pathways?

Some minor questions:

Network structure: the M non-overlapping subpopulations are hardcoded into the exc neurons rnn? This merely means that they receive the same input, but the connectivity is still random?

External inputs: how long in time are individual elements? When for example sequence ABCD is given, does each letter trigger a single spike? What if it triggers multiple? I suppose the refractory period does not allow this.

Equation 1: x_j is written after the summation and y_i before. For esthetic reasons I would put both either before or after.

Line 223: why are the minimal permanences uniformly distributed? Is this how they are initialized, and hence guarantee the necessary heterogeneity?

Are the parameters, for example dAP threshold of 59pA etc, taken from a particular previous study or finetuned?

What is the purpose of the recurrent EE connectivity? Am I correct in saying that the feedforward pathways, being carved out, is what makes the model work?

**Have the authors made all data and (if applicable) computational code underlying the findings in their manuscript fully available?**

Reviewer #1: Yes

Reviewer #2: Yes

PLOS authors have the option to publish the peer review history of their article (what does this mean?). If published, this will include your full peer review and any attached files.

Reviewer #1: No

Reviewer #2: No
---

## [Decision Letter · Decision Letter 1]

7 Mar 2022

Dear Mr Bouhadjar,

Thank you very much for submitting your manuscript "Sequence learning, prediction, and replay in networks of spiking neurons" for consideration at PLOS Computational Biology.

In light of the reviews (below this email), we would like to invite the resubmission of a significantly-revised version that takes into account the reviewers' comments.

As you can see, the remaining reviewer made the point that there is a gap between your model features and key features of experimentally observed neuronal activity. I concur with the reviewer that although your efforts to implement the HTM in a spiking system are laudable, from the current results  it not clear how helpful the model may be in the future for experimental studies. This is important because the scope of PLOS Computational Biology is aimed towards biological discovery through modelling. Comments in this direction were made by the reviewer in the first rounds of review and as far as I can see you added text to the discussion but didn't consider the reviewer's suggestions towards adding simulations. Therefore, I strongly recommend that you address these comments about suggested further simulations, e.g. to test the model with some background activity and rate modulation of the input.

We cannot make any decision about publication until we have seen the revised manuscript and your response to the reviewers' comments. Your revised manuscript is also likely to be sent to reviewers for further evaluation.

Sincerely,

Stefan Kiebel

Associate Editor

PLOS Computational Biology

Samuel Gershman

Deputy Editor

PLOS Computational Biology

Reviewer's Responses to Questions

**Comments to the Authors:**

Reviewer #1: The authors propose a new implementation of the Hierarchical Temporal Memory (HTM) algorithm in terms of biologically interpretable neural networks. The model has some interesting and valuable ideas but some of the simplifications applied in the simulations and model limit in my eyes its impact.

As outlined in the major points of my previous review, the activity produced by the model is extremely far from what is observed experimentally. Input stimuli are passed to the network by one single spike; there is no rate modulation, neither in the input signal nor in the response of the network; and the model does not display any background activity. These are extreme conditions, and leave the reader wondering how the model would function in a more realistic scenario. Therefore, I had suggested new simulations.

For example, a first quick way to address some of these concerns could have been to re-run the existing model (learning and prediction) with input signals that include (beyond the sequences to learn) also noise spikes and/or rate modulation. Such analyses are quick to perform, as they do not require any modification in the model, and their outcome, either positive or negative, would be highly informative.

The authors did not perform new analyses but addressed the concerns by adding a paragraph in the Discussion of the manuscript (and a new Suppl. Fig. 5) where they discussed the degree of synchronization present in experimental data and how synchrony might go unnoticed when subsampling.

In their reply letter the Authors write: “In a new paragraph added to “Discussion: Limitations and outlook”, we point at the apparent mismatch between the firing statistics in our model and those found in nature, and explain how a more natural irregular activation of sequences, parallel processing, and subsampling would resolve this mismatch”. In my opinion, this does not resolve the mismatch and does not address the concern raised. Moreover, this line of argumentation (together with the addition of Suppl. Fig. 5) appears to suggest that brain activity 'is' exclusively composed of synfire chains. While there is large experimental evidence of the presence of highly synchronized neuronal ensemble in the brain, assuming that such kind of activity constitutes the totality of the neuronal activity is a very strong claim.

Therefore, in my opinion it is necessary to perform additional simulations, on the line of those described above, to provide insightful addition to the field suited for PLOS Computational Biology.

**Have the authors made all data and (if applicable) computational code underlying the findings in their manuscript fully available?**

Reviewer #1: Yes

PLOS authors have the option to publish the peer review history of their article (what does this mean?). If published, this will include your full peer review and any attached files.

Reviewer #1: No
---

## [Decision Letter · Decision Letter 2]

19 May 2022

Dear Mr Bouhadjar,

Thank you very much for submitting your manuscript "Sequence learning, prediction, and replay in networks of spiking neurons" for consideration at PLOS Computational Biology. As with all papers reviewed by the journal, your manuscript was reviewed by members of the editorial board and by several independent reviewers. The reviewers appreciated the attention to an important topic. Based on the reviews, we are likely to accept this manuscript for publication, providing that you modify the manuscript according to the review recommendations.

Sincerely,

Stefan Kiebel

Associate Editor

PLOS Computational Biology

Samuel Gershman

Deputy Editor

PLOS Computational Biology

[LINK]

Reviewer's Responses to Questions

**Comments to the Authors:**

Reviewer #1: I appreciate the new simulations performed by the Authors, which can better inform the readers about the response range of the model.

I would suggest the publication of the current paper pending one final change.

On lines 639-640 the authors state: ‘As shown in Fig. S6 and Fig. S7, the spiking activity and the prediction performance of the TM model are robust with respect to low and moderate levels of synaptic background activity'.

In figure S6 are shown results for a no-noise condition (panel A) and for two different degrees of noise (panels B and C). In figure S7 it is shown that for the second of these two noise conditions the network loses its ability to learn sequences.

Given these results, I would kindly ask the authors to remove from the previous sentence the words “and moderate”.

Firstly, only two levels of noise have been tested and for only one of these two the network maintained the desired qualities. Hence, it is not clear to what refers the sentence ‘robust with respect to low and moderate levels of synaptic background activity’. Secondly, the model works only until 'non-task related background spikes' are generated (as confirmed by the authors), which is over all a clearly very low noise condition (as also visible in Fig. S6 B).

**Have the authors made all data and (if applicable) computational code underlying the findings in their manuscript fully available?**

Reviewer #1: Yes

PLOS authors have the option to publish the peer review history of their article (what does this mean?). If published, this will include your full peer review and any attached files.

Reviewer #1: No

Figure Files:

Data Requirements:

Reproducibility:

References:

---

## [Editor Report · Decision Letter 3]

20 May 2022

Dear Mr Bouhadjar,

We are pleased to inform you that your manuscript 'Sequence learning, prediction, and replay in networks of spiking neurons' has been provisionally accepted for publication in PLOS Computational Biology.

Best regards,

Stefan Kiebel

Associate Editor

PLOS Computational Biology

Samuel Gershman

Deputy Editor

PLOS Computational Biology

---

## [Editor Report · Acceptance letter]

15 Jun 2022

PCOMPBIOL-D-21-02013R3 

Sequence learning, prediction, and replay in networks of spiking neurons

Dear Dr Bouhadjar,

I am pleased to inform you that your manuscript has been formally accepted for publication in PLOS Computational Biology. Your manuscript is now with our production department and you will be notified of the publication date in due course.

With kind regards,

Livia Horvath
